# Needle In A Video Haystack: A Scalable Synthetic Evaluator for Video MLLMs

**Zijia Zhao**[1,2*]**, Haoyu Lu**[3*]**, Yuqi Huo**[4]**, Yifan Du**[3]**, Tongtian Yue**[1,2]**,**
**Longteng Guo**[1,2]**, Bingning Wang**[4]**, Weipeng Chen**[4]**, Jing Liu**[1,2†]

[1]Institute of Automation, Chinese Academy of Sciences
[2]School of Artificial Intelligence, University of Chinese Academy of Sciences
[3]Gaoling School of Artificial Intelligence, Renmin University of China
[4]Baichuan Inc.

## Abstract

Video understanding is a crucial next step for multimodal large language models (MLLMs). Various benchmarks are introduced for better evaluating the MLLMs. Nevertheless, current video benchmarks are still inefficient for evaluating video models during iterative development due to the high cost of constructing datasets and the difficulty in isolating specific skills. In this paper, we propose **Video-NIAH** (**Video N**eedle **I**n **A H**aystack), a benchmark construction framework through synthetic video generation. VideoNIAH decouples video content from their query-responses by inserting unrelated visual 'needles' into original videos. The framework automates the generation of query-response pairs using predefined rules, minimizing manual labor. The queries focus on specific aspects of video understanding, enabling more skill-specific evaluations. The separation between video content and the queries also allow for increased video variety and evaluations across different lengths. Utilizing VideoNIAH, we compile a video benchmark, **VNBench**, which includes tasks such as retrieval, ordering, and counting to evaluate three key aspects of video understanding: temporal perception, chronological ordering, and spatio-temporal coherence. We conduct a comprehensive evaluation of both proprietary and open-source models, uncovering significant differences in their video understanding capabilities across various tasks. Additionally, we perform an in-depth analysis of the test results and model configurations. Based on these findings, we provide some advice for improving video MLLM training, offering valuable insights to guide future research and model development.

## 1 Introduction

Multimodal Large Language Models (MLLMs) (Li et al., 2023d; Liu et al., 2023; Lu et al., 2024; OpenAI, 2023; Team et al., 2023; Wang et al., 2023; Liu et al., 2024b) have recently made significant strides in understanding visual content. Video understanding is one of the most crucial next steps to mirror real-world scenarios, and numerous video-centric MLLMs (Li et al., 2023e; Maaz et al., 2023; Luo et al., 2023; Lin et al., 2023; Reid et al., 2024) have been proposed. Recent research has further established video benchmarks (Li et al., 2023f; Ning et al., 2023; Chen et al., 2023; Liu et al., 2024e) to assess specific aspects of video understanding in these models.

Most existing video benchmarks are designed to assess the general understanding capabilities of video models and are curated to reflect real-world data. However, we argue that these benchmarks are inefficient for evaluating video models during the iterative development process. The inefficiency arises from two main issues: the high cost of constructing quality datasets and the inability to decouple different aspects of video comprehension, which makes it challenging to pinpoint specific weaknesses in the models.

The first challenge is the construction of high-quality, realistic video benchmarks, a process that remains time-consuming and complex, as illustrated in Fig. 1(a). Selecting videos based on targeted capabilities is crucial (Mangalam et al., 2024; Yu et al., 2019). Additionally, creating these benchmarks often requires labor-intensive tasks such as prompt engineering (Li et al., 2023f; Ning et al.,

---

*Equal contribution.
†Corresponding author.

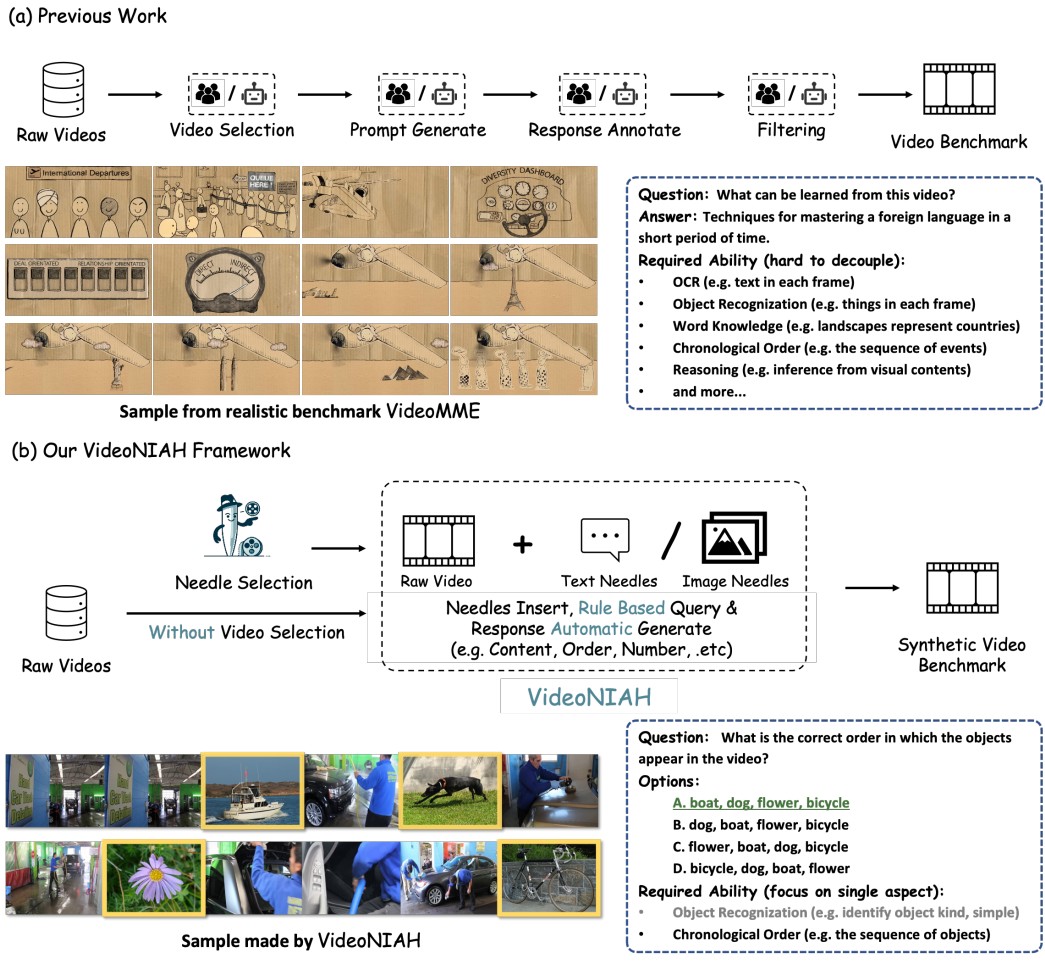

Figure 1: Comparison between **VideoNIAH** and *de facto* paradigm. Previous methods require extensive model/manual design to construct real-world data, and the resulting question-answer pairs often assess multiple video understanding capabilities simultaneously, making it difficult to decouple specific skills. In contrast, **VideoNIAH** framework automates data construction through predefined rules. These rules correspond to distinct aspects of video understanding, enabling the precise identification of weaknesses in a specific capabilities. We show detailed rules in Appendix A.4

2023; Chen et al., 2023; Liu et al., 2024e), manual annotation (Jang et al., 2017; Mangalam et al., 2024; Ning et al., 2023; Yu et al., 2019; Chen et al., 2023), and data filtering (Xiao et al., 2021; Li et al., 2023f; Liu et al., 2024e; Yu et al., 2019) to ensure accurate alignment between query-answer pairs and video content. Furthermore, the risk of data leakage arises when using query-response pairs derived from real-world videos, as these videos may have been previously used during model training, compromising the fairness and objectivity of benchmark evaluations.

The second limitation of existing benchmarks is their comprehensive nature (Fu et al., 2024; Zhou et al., 2024; He et al., 2024; Liu et al., 2024e; Du et al., 2024; Wu et al., 2024), which requires models to address multiple aspects of video content simultaneously. For instance, as shown in Fig. 1(a), the example from VideoMME (Fu et al., 2024) benchmark demands a broad range of capabilities, including Optical Character Recognition (OCR), object detection, word knowledge, chronological order and temporal reasoning, *etc*. This comprehensive nature of such benchmarks makes it difficult to evaluate specific weaknesses in any single capability, complicating the skill-specific improvement during the model iteration process.

To address these challenges, we propose **VideoNIAH** (**Video N**eedle **I**n **A H**aystack), a novel and scalable framework for constructing video benchmarks using synthetic video generation. This approach is inspired by advancements in language model evaluations (Kamradt, 2023; Song et al.,

2024; Hsieh et al., 2024). VideoNIAH introduces an innovative method that decouples test videos from their corresponding query-response pairs by embedding unrelated image or text "needles" into the original video "haystacks". This technique enables the use of diverse video sources with flexible lengths, offering significant scalability and adaptability. Moreover, VideoNIAH allows for the automated design of video understanding probing tasks by inserting multiple spatio-temporal "needles" into videos and generating the corresponding query-response pairs based on predefined rules. These rules are tailored to probe specific aspects of video understanding, ensuring that the evaluation of one capability is minimally influenced by other abilities. This framework significantly reduces the need for human labor while providing precise assessments of targeted model skills.

Utilizing VideoNIAH, we compile a decoupled video benchmark, **VNBench**, which includes tasks such as retrieval, ordering, and counting. These three tasks independently point to the three most important aspects in video understanding: temporal perception, identify chronological order and understanding spatio-temporal coherence. By isolating these abilities, VNBench enables a more focused evaluation of specific video comprehension capabilities, facilitating a clearer assessment of model performance across distinct dimensions. Additionally, since the video sampled in VideoNIAH method is decoupled with task-specific query-response pairs, we can add various videos in any length and domain into the test set, which makes video evaluation under any context length possible. We evaluate 12 video understanding MLLMs on VNbench, including 3 proprietary models and 9 open-source models. We observe a significant performance gap between the proprietary and open-source models on temporal tasks (retrieval and ordering), with the proprietary models showing clear advantages. Moreover, most models perform poorly in spatial-temporal tasks (counting), suggesting that current video models are still far from perfect. Leveraging the flexibility of the VideoNIAH framework, we further investigate the impact of various components within VNBench, including video context length, the number, position, and characteristics of inserted needles. Moreover, we examine the effects of video model training settings and offer valuable advice for improving video MLLM training practices. Our contributions are summarized as follows:

- We proposed **VideoNIAH**, a simple, flexible and scalable synthetic framework for designing skill-targeted video benchmark, which can be used to explore the different aspects of video understanding without heavy annotation resource cost.

- To the best of our knowledge, we are the first to propose a synthetic video benchmark **VNBench**, which thoroughly examines three import aspects in video understanding: retrieval, ordering and counting, on a board range of context length.

- We conduct a thorough analysis of both proprietary and open-source video models on VNBench, providing insights into the effects of context length, needle number, type, and position on model performance. Additionally, we offer practical advice for improving video MLLM training practices on a comprehensive model analysis, with the aim of inspiring future research.

## 2 RELATED WORKS

### 2.1 VIDEO MLLMS AND VIDEO BENCHMARKS

Recently, various multimodal large language models (MLLMs) have been developed for video understanding. VideoChat (Li et al., 2023e) and Video-LLaMA (Zhang et al., 2023) are pioneering efforts that utilize large language models (LLMs) for this purpose. Subsequent models (Lin et al., 2023; Li et al., 2023g;f; Liu et al., 2024d; Zhang et al., 2024), following training strategies similar to those of Valley (Luo et al., 2023) and VideoChatGPT (Maaz et al., 2023), have been developed using open-source video data. MovieChat (Song et al., 2023) innovates by designing a complex mechanism that incorporates both short-term and long-term memory, enhancing Video-LLaMA's capability for understanding extended videos. Several video MLLMs (Wang et al., 2024b; Li et al., 2024; Zhao et al., 2023) apply more video SFT data to achieve more comprehensive ability or integrate external modality information to enhance performance. Meanwhile, proprietary models like GPT-4V (OpenAI, 2023) and Gemini 1.5 Pro (Reid et al., 2024) demonstrate superior video understanding capabilities through larger model parameters and more comprehensive training procedures.

In order to evaluate the video understanding capability of these MLLMs, several video benchmarks have been proposed. Moving beyond traditional video question-answering (QA) datasets (Xu et al.,

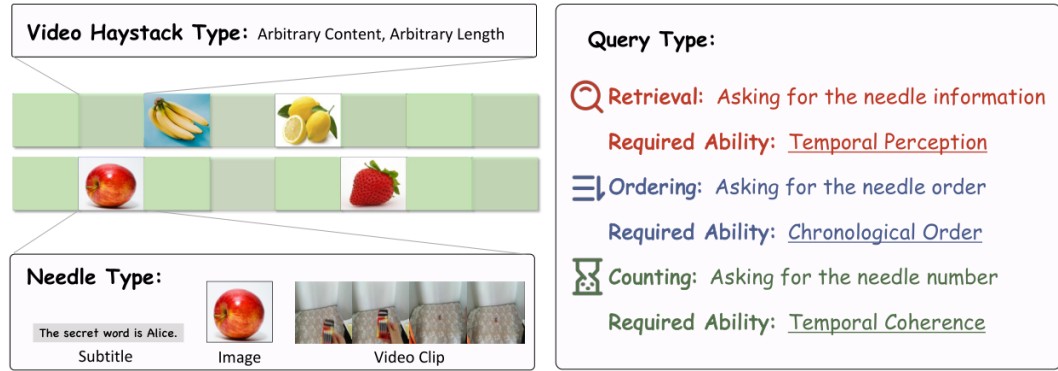

Figure 2: Construction Overview of VNBench within the VideoNIAH Framework. Each sample in VNBench is composed of several inserted needles, query-response pairs generated by predefined rules, and a randomly selected video haystack.

2016; Caba Heilbron et al., 2015; Chen & Dolan, 2011; Grunde-McLaughlin et al., 2021; Jang et al., 2017; Liu et al., 2018), more comprehensive benchmarks (Li et al., 2023f; Maaz et al., 2023; Liu et al., 2024e; Chen et al., 2023; Ning et al., 2023; Fu et al., 2024; Zhou et al., 2024; Du et al., 2024) have been proposed to encapsulate the diverse characteristics of video data comprehensively. Additionally, specific benchmarks (Patraucean et al., 2024; Li et al., 2023c; Mangalam et al., 2024; Xiao et al., 2021; Song et al., 2023) have been designed to evaluate models' proficiency in understanding long-context videos within a QA framework. However, these video benchmarks, which are based on realistic videos, may encounter data leakage issues and require laborious annotation processes. VideoNIAH is a scalable synthetic video benchmark, that avoids the above issues and evaluates different video understanding abilities on a boarder range of context lengths.

## 2.2 Synthetic Benchmarks

Synthetic benchmarks (Kamradt, 2023; Li et al., 2023b; Liu et al., 2024a; Reid et al., 2024) provide more control over factors like sequence length and task complexity. They are largely unaffected by the parametric knowledge acquired during model training, thereby eliminating the risk of data leakage. Needle in a Haystack (Kamradt, 2023) (NIAH) first introduces a synthetic framework for evaluating the in-context retrieval capabilities of long-context large language models (LLMs). This method involves embedding a random synthetic statement within an unrelated long context and subsequently querying the model to retrieve this statement. Other approaches (Mohtashami & Jaggi, 2023) utilize special tokens or strategies to develop synthetic benchmarks tailored for LLMs. Counting-stars (Song et al., 2024) and RULER (Hsieh et al., 2024) enhance the original 'needle in a haystack' task by introducing more complex settings and emphasizing the long-range dependencies of the inserted statements. VideoNIAH represents the first holistic synthetic benchmark for video understanding with diverse needle types, various query formats and long context length.

## 3 Evaluation Data Recipe

### 3.1 VideoNIAH: A Flexible Synthetic Framework

VideoNIAH is a synthetic framework to assess video model comprehension inspired by the "needle in a haystack (NIAH)" test (Kamradt, 2023). Each sample in VideoNIAH involves "needles" representing inserted information, "haystack" for the original video context, and "query" directing the extraction of needles. The "needle" information in VideoNIAH is independent of the video content in "haystack". This independence necessitates a focus on the synthetic design of the "needle" and the rule to generate specific query-response pairs to accurately assess video understanding capabilities.

We provide a comprehensive VideoNIAH recipe tailored for *the natural characteristics of videos* considering the following reasons. **1)** The inherent spatio-temporal relationships in videos motivate the use of both intra-frame and inter-frame "needles". We employ two strategies to construct these spatio-temporal "needles": editing within individual video frames and inserting visual content across

multiple frames. These needles can take the form of textual subtitles, static images, dynamic video clips, *etc*. **2)** Multiple segments should be captured in a video at once. Consequently, we advocate for an increase in the number of "needles" from one to several in VideoNIAH. **3)** The presence of extensive long-range associations in lengthy videos underscores the need to introduce more challenging queries that depend on long-term dependencies. In conclusion, the VideoNIAH synthetic framework is characterized by its combination of spatial and temporal analysis, an increased number of needles, and the introduction of long-dependency queries.

### 3.2 VNBench: A Basic Video Understanding Benchmark

We construct a video understanding benchmark VNBench with the above synthetic method, which is shown in Fig. 2. VNBench contains three distinct tasks to address various aspects of video understanding, including a short-dependency task (retrieval) and two long-dependency tasks (ordering and counting). Additionally, each task incorporates various types of "needles", considering both intra-frame editing and inter-frame inserting methods to enrich the comprehensiveness of the evaluation.

**Retrieval** task is the basic NIAH task aimed at evaluating the long-context video model's ability to retrieve a single needle. This task specifically assesses the model's capability for **temporal perception** and understanding across the time dimension. The retrieval task in VNBench is categorized into two types based on the nature of the "needle": intra-frame editing needles and inter-frame inserting needles. For intra-frame editing, we use inserted subtitles as the needle, while for inter-frame insertion, static images are employed. Additionally, the inter-frame inserting needles are further divided into two difficulty levels: simple and hard images.

**Ordering** task is more challenging compared to the retrieving task mentioned above. In the ordering task, the model needs to identify the correct **chronological order** of all inserted needles. Ordering demonstrates the ability to comprehend temporal dynamics and sequence events accurately in video, which reflects the model's proficiency in temporal reasoning. Similar to the retrieval task, the ordering task in VNBench is divided into two sub-tasks based on needle type: intra-frame editing needles and inter-frame inserting needles. And the inter-frame inserting needles are also further divided into two difficulty levels.

**Counting** is also a challenging task for long-context video models. In the counting task, the model is required to provide the number of times the specified object appears in a video. Successfully counting recurring content over an extended video demonstrates the model's ability to recognize and track **spatio-temporal coherence** patterns, which is closely tied to its proficiency in maintaining temporal consistency and an internal representation of identified elements across different segments of the video. The counting task in VNBench is also divided into two sub-tasks: intra-frame editing needles and inter-frame inserting needles. Unlike the previous tasks, the counting task's difficulty is divided into two levels within the editing sub-task: subtitles and local image regions. In counting-E2 task, the local image regions are applied to partial areas of the original video frames, testing the model's ability to recognize spatio-temporal coherence and maintain consistency over time.

Table 1: Task Statistics in VNBench

| Task Name | Needle Type | Needle Position | | #Sample |
| | | Edit Region | Insert Frame | |
| --- | --- | --- | --- | --- |
| *Short-dependency Ability* | | | | |
| Retrieval-E | Single Subtitle | ✓ | | 150 |
| Retrieval-I1 | Single Fruit Image | | ✓ | 150 |
| Retrieval-I2 | Single Landmark Image | | ✓ | 150 |
| *Long-dependency Ability* | | | | |
| Ordering-E | Four Subtitle | ✓ | | 150 |
| Ordering-I1 | Four Fruit Image | | ✓ | 150 |
| Ordering-I2 | Four Object Image | | ✓ | 150 |
| Counting-E1 | Multiple Subtitle | ✓ | | 150 |
| Counting-E2 | Multiple Object Image | ✓ | | 150 |
| Counting-I | Multiple Object Image | | ✓ | 150 |
| Total | | | | 1350 |

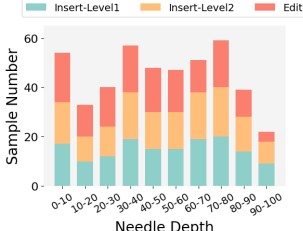
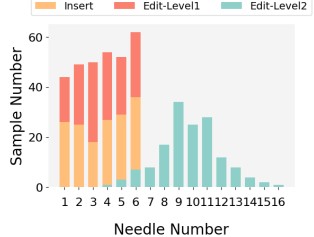

Figure 3: Needle Depth Distribution in Retrieval task

Figure 4: Needle Number Distribution in Counting task

As shown in Table 1, the VNBench benchmark contains three tasks, each with three sub-tasks focusing on different aspects. We sampled 150 video *haystacks* from three video data sources: MSRVTT (Xu et al., 2016), NeXT Videos (Xiao et al., 2021), and ActivityNet (Caba Heilbron et al., 2015). These videos contain various scenarios and range from 10 to 180 seconds in duration. We create nine task samples for each video haystack, resulting in a total of 1350 samples for the entire test set. Additionally, we have created an extremely long video test set called VNBench-Long*, comprising four tasks, where the videos range from 10 to 30 minutes in length. Each sample consists of a video with inserted needles, a question, four answer options, and one ground-truth answer. The duration of each needle is set to 1 second. For the retrieval task, the answer options are sampled from the entire set of ground-truth candidates. For the ordering task, the negative answer options are shuffled versions of the ground-truth. For the counting task, we sample three negative options from a normal distribution centered around the ground-truth number. Further details on the construction of VNBench can be found in Appendix A.

## 3.3 AUTOMATIC FILTER IN NEEDLE SELECTION

To prevent confusion between needle images and the video haystacks, we employ a straightforward yet effective filtering method for needle selection. We adopt a CLIP[†] model to compute the image similarity between potential needle images and frames extracted from the video haystack. If the maximum similarity exceeds a predefined threshold (0.7, in VNBench), the candidate needle images are discarded from the current round of data generation.

## 3.4 EVALUATION STRATEGY

We define all tasks as multiple-choice questions. For each query, we provide four choices, with only one being correct. To reduce randomness in multiple-choice questions, we adopt a circular evaluation strategy. Each sample is evaluated four times, with the options shuffled each time. A sample is considered correctly answered only if the model selects the correct option all four times. We report the accuracy of different tasks.

## 4 EVALUATION RESULTS

### 4.1 MODELS

We select 12 video understanding models, including 9 open-source models and 3 proprietary models. The proprietary models are Gemini 1.5 Pro (Reid et al., 2024) and the GPT-4 series (OpenAI, 2023), accessed via the official API. We evaluate 2 different GPT-4 version include `gpt-4-turbo-2024-04-09` and `gpt-4o-2024-05-13`. The open-source video MLLMs include LLaVA-NeXT-Video (Zhang et al., 2024), ST-LLM (Liu et al., 2024d), LLaMA-VID (Li et al., 2023g), Video-LLaVA (Lin et al., 2023), VideoChatGPT (Maaz et al., 2023), VideoChat2 (Li et al., 2023f), Video-LLaMA2 (Zhang et al., 2023), Qwen2-VL (Wang et al., 2024b) and LLaVA-OneVision (Li et al., 2024). The detailed model setting is shown in Appendix C.1.

### 4.2 MAIN RESULTS

In Table 11, we report the whole result on 9 VNBench tasks and the overall score for each model on the main split of VNBench. We summarize the result as follows:

**1) Proprietary models perform better than open-source models on most VNBench tasks.** In terms of overall accuracy, the highest performance among open-source models (58.7% for LLaVA-OneVision-72B) and the highest performance among proprietary models (66.7% for Gemini 1.5 Pro) differ by 8.0%. Additionally, the average accuracy of proprietary models is also significantly higher than that of open-source models.

**2) Performance on multiple-needle long-dependency tasks is lower than on single-needle short-dependency tasks.** When comparing the accuracy across different tasks, we find that most models

---

*The results are provided in the Appendix D.

[†]openai/clip-vit-base-patch32

Table 2: Evaluation Results on VNBench. VNBench comprises three synthetic tasks constructed using the VideoNIAH method, with each task divided into three splits. "E" denotes intra-frame editing needles, while "I" represents inter-frame inserting needles. The numbers "1" and "2" refer to the difficulty levels of the sub-tasks, with "1" indicating simple and "2" indicating hard. In total, we evaluated 3 proprietary models and 9 open-source models across these tasks.

| Video MLLMs | Retrieval | | | | Ordering | | | | Counting | | | | Overall |
|---|---|---|---|---|---|---|---|---|---|---|---|---|---|
| | E | I-1 | I-2 | Avg. | E | I-1 | I-2 | Avg. | E-1 | E-2 | I | Avg. | |
| *Proprietary Models* | | | | | | | | | | | | | |
| Gemini 1.5 Pro (Reid et al., 2024) | 100.0 | 96.0 | 76.0 | 90.7 | 90.7 | 95.3 | 32.7 | 72.9 | 60.7 | 7.3 | 42.0 | 36.7 | 66.7 |
| GPT-4o (OpenAI, 2023) | 100.0 | 98.0 | 87.3 | 95.3 | 88.4 | 86.6 | 45.2 | 73.4 | 36.8 | 0.0 | 36.1 | 24.5 | 64.4 |
| GPT-4-turbo (OpenAI, 2023) | 100.0 | 99.3 | 82.0 | 93.7 | 42.6 | 22.8 | 23.0 | 29.5 | 37.6 | 0.0 | 32.4 | 23.3 | 48.9 |
| *Open-source MLLMs* | | | | | | | | | | | | | |
| VideoChatGPT (Maaz et al., 2023) | 4.7 | 4.7 | 0.7 | 3.3 | 2.7 | 11.3 | 0.0 | 4.7 | 2.0 | 4.0 | 6.7 | 4.2 | 4.1 |
| Video-LLaMA2 (Zhang et al., 2023) | 1.2 | 26.0 | 6.0 | 11.1 | 0.0 | 0.0 | 0.0 | 0.0 | 2.0 | 4.7 | 0.7 | 2.4 | 4.5 |
| LLaMA-VID-7B (Li et al., 2023g) | 28.0 | 28.0 | 19.3 | 25.1 | 0.7 | 0.0 | 0.0 | 0.2 | 4.0 | 2.7 | 14.7 | 7.1 | 10.8 |
| Video-LLaVA-7B (Lin et al., 2023) | 26.0 | 28.0 | 17.3 | 23.8 | 0.7 | 0.7 | 2.0 | 1.1 | 16.7 | 0.7 | 20.0 | 12.4 | 12.4 |
| VideoChat2 (Li et al., 2023f) | 43.4 | 40.0 | 14.6 | 32.7 | 0.0 | 0.0 | 1.3 | 0.4 | 3.3 | 0.7 | 8.0 | 4.0 | 12.4 |
| LLaVA-NeXT-Video-7B (Zhang et al., 2024) | 56.7 | 56.7 | 19.3 | 44.2 | 0.7 | 0.0 | 0.7 | 0.4 | 6.7 | 14.6 | 25.3 | 15.5 | 20.1 |
| ST-LLM (Liu et al., 2024d) | 58.0 | 64.7 | 31.3 | 51.3 | 0.0 | 0.0 | 0.0 | 0.0 | 21.3 | 1.3 | 27.3 | 16.7 | 22.7 |
| LLaVA-OneVision-0.5B (Li et al., 2024) | 88.7 | 80.7 | 22.0 | 63.8 | 2.7 | 1.3 | 2.7 | 2.2 | 6.7 | 9.3 | 18.7 | 11.6 | 25.9 |
| Qwen2-VL-7B (Wang et al., 2024b) | 98.0 | 76.0 | 33.3 | 69.1 | 16.0 | 12.7 | 8.7 | 12.4 | 26.0 | 9.3 | 24.7 | 20.0 | 33.9 |
| LLaVA-OneVision-7B (Li et al., 2024) | 88.7 | 87.3 | 55.3 | 77.1 | 70.0 | 50.0 | 37.3 | 52.4 | 41.3 | 8.7 | 27.3 | 25.8 | 51.8 |
| LLaVA-OneVision-72B (Li et al., 2024) | 90.7 | 86.7 | 57.3 | 78.2 | 78.0 | 74.0 | 54.0 | 68.7 | 42.7 | 14.7 | 30.7 | 29.3 | 58.7 |

perform much better on retrieval tasks than on ordering and counting. For most proprietary models, they can retrieve almost all the inserted information (for example, 100% accuracy on Retrieval-E task). For some open-source models (such as ST-LLM, LLaVA-NeXT-Video, Qwen2-VL, LLaVA-OneVision), the retrieval accuracy is also significantly higher than on the other two tasks.

**3) The gap between open and proprietary models in the ordering task is enormous.** The most advanced proprietary models are far ahead of other models in the ordering task (with Gemini 1.5 Pro at 72.9% accuracy on ordering task and GPT-4o at 73.4%), while most open-source models are nearly incapable of completing the ordering task with the exception of the LLaVA-OneVision series. This may be due to most open-source models' training processes overlooking the modeling of temporal sequences, thereby impairing the models' ability to process temporal relationships.

**4) Counting is difficult, especially when counting hard-to-identify needles.** Even on the more advanced proprietary models, the performance in counting is not very good. Moreover, on the Counting-E-2 task (detecting and tracking information deeply embedded within specific spatial areas of video segments), all models perform poorly, suggesting that current video models still lack the capability to deeply understand and model the fine-grained spatial-temporal relationships in videos.

## 5 RESULT ANALYSIS

In this section, we further analyze the evaluation results on VNbench, conducting an in-depth analysis of different video understanding capabilities.

**Effect of Haystack Length** Since the query-response pairs constructed through VideoNIAH are unrelated to the original video content, we can fairly compare the models' video understanding ability to handle videos of different lengths by dividing them according to the length of the sample videos. In Fig. 5, we divide the VNBench data into three splits based on the video haystack duration: short (10-30s), medium (30-60s), and long (60-180s). We observe that as the duration length of the videos processed changes, the performance of the proprietary models does not fluctuate significantly, thanks to their longer context processing windows (128k tokens for GPT-4 and 1M tokens for Gemini 1.5 Pro). However, for open-source models, handling longer-duration videos proves difficult, and models such as VideoChat2, LLaVA-NeXT-Video, and ST-LLM show a significant performance decline. This indicates that current models are still limited in the duration of video they can effectively handle.

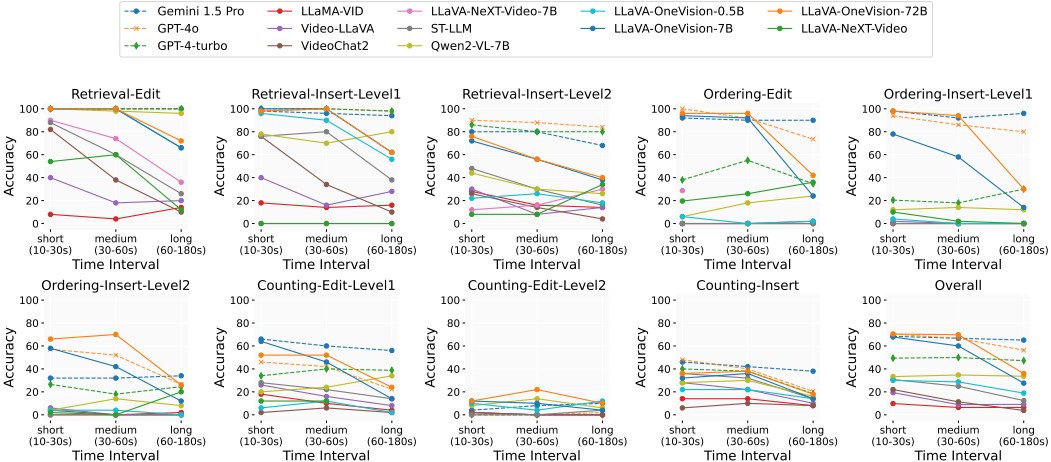

Figure 5: Task performance on different video durations. We divide all VNBench videos into 3 splits: short(10-30s), medium(30-60s) and long(60-180s). All the numerical results are in the Table 11.

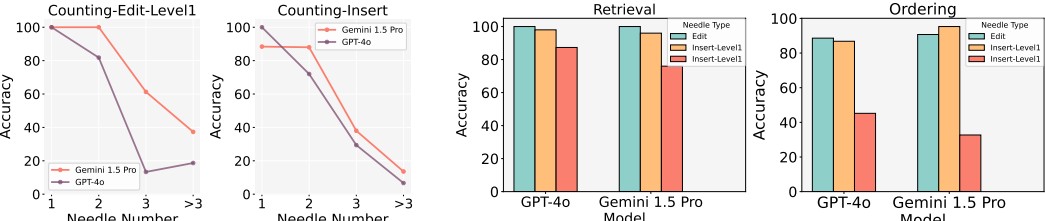

Figure 6: Effect of needle number in VNBench-Counting task.

Figure 7: Effect of needle type in VNBench-Retrieval and VNBench-Ordering task.

**Effect of Needle Number** In Fig. 6, we show the effect of the needle number in counting tasks, where the number of inserted needles varies among different samples. We observe that as the number of needles increases, the model's performance in the counting task significantly declines. This result highlights deficiencies in video understanding models regarding tracking and memory of objects within video content. It indicates that current video understanding models still need further optimization and improvement in understanding spatio-temporal relationships, attention mechanisms, and long-term memory processing.

**Effect of Recognizing Ability** Visual recognition is a fundamental ability for video MLLMs, , as comprehensively analyzing visual contents on all frames is essential to understand their interrelationships. In Fig. 7, we illustrate the impact of different needle types on the retrieval task, highlighting the importance of robust visual recognition skills by comparing various needle categories. The retrieval task encompasses different sub-tasks, each probing a distinct aspect of visual recognition. The Retrieval-E sub-task evaluates the model's ability to identify specific local patches. The Retrieval-I-1 sub-task involves recognizing common objects. Conversely, the Retrieval-I-2 sub-task demands extensive world knowledge and a keen eye for detail to identify landmark images, challenging the model's fine-grained visual recognition capabilities. We note that proprietary models like Gemini 1.5 Pro and GPT-4o excel at recognizing subtitles and common images. However, more complex images tend to introduce confusion within the video, leading to a noticeable drop in accuracy in the Retrieval-I-2 sub-task compared to Retrieval-I-1. This variation highlights the necessity of strong visual recognition abilities for advanced video MLLMs.

**Effect of Needle Position** We also explore the effect of needle position in Fig. 8. We fix the video haystack and query-response pair in this position test on Retrieval-I-1 task, just modifying the haystack length and needle position. The video haystack varies from 10 to 180 seconds, while the needle is inserted at depths ranging from 0% to 90%. We evaluate the average accuracy for each position using 32 different VNBench-Retrieval-I-1 samples, varying the needle depth and haystack length. For each sample, haystacks of different lengths are randomly cut from the same long video.

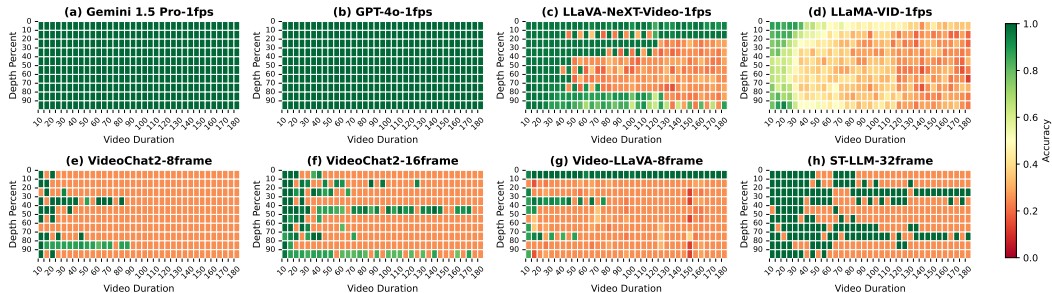

Figure 8: Results of varying depth and context length in VNBench-Retrieval-I-1. The x-axis represents the video duration, while the y-axis indicates the context depth where the needle resides. Different color indicates different average accuracy of 32 different samples.

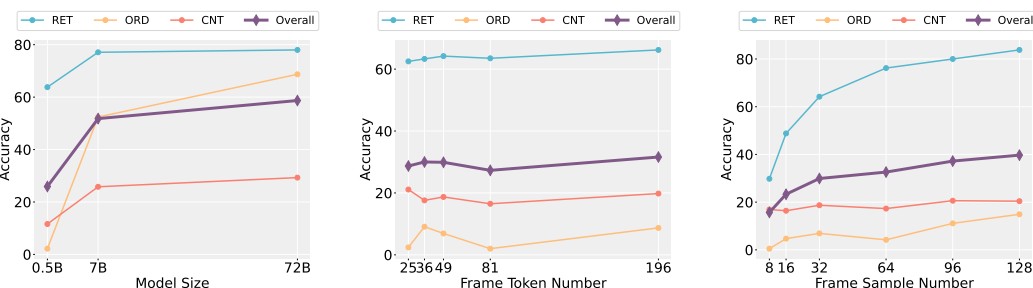

Figure 9: Model analysis on model size. We evaluate LLaVA-OneVision model family with different model sizes.

Figure 10: Model analysis on token number per frame. All models are trained with 32 frames.

Figure 11: Model analysis on frame number. All models are trained with different frames and 49 frame token numbers.

We observe three distinct patterns in this test. For the proprietary models, the test intervals we used are much shorter than their context window lengths, allowing these models to precisely recall all inserted information (Fig. 8 a,b).

For open-source models using sequential sampling, such as LLaMA-VID and LLaVA-NeXT-Video, a phenomenon similar to 'lost in the middle' (Liu et al., 2024c) in language models is evident (Fig. 8 c,d), where the models tend to recall information at the beginning and end of long sequences, but pay less attention to the middle. For open-source models that employ a uniform sampling strategy, such as Video-LLaVA and VideoChat2, they uniformly sample the input video to ensure a fixed number of frames are fed into the model. Thus, their recall results in position tests are more closely related to the sampling strategy, displaying a similar "bar-shaped" phenomenon (Fig. 8 e-h), where the successful recall of a needle is tightly linked to whether it was sampled.

## 6 MODEL ANALYSIS

In this section, we mainly analyze several important items we think may influence the video MLLM's ability on spatial and temporal understanding. We use VNBench to validate the influence of these model settings. And in all experiments in this section, the training recipe of our model is kept same. We list the detailed training recipe in Appendix G.

**Model Size** We studied the impact of model parameters on the LLaVA-OneVision (Li et al., 2024) family, with a particular focus on the parameters related to the language model. In Fig. 9, we find that as model parameters increase, performance across all VNBench sub-tasks improves, with the ordering task showing the most significant gains. This suggests that enhanced language abilities strengthen the model's ability to perceive chronological order. However, we observed only marginal improvement in the coherence task (counting), which aligns with the well-known difficulty language models face with counting tasks.

**Token Number Per Frame** We explore the impact of varying the number of tokens per frame. Intuitively, increasing the token density per video frame should enhance spatial understanding. To investigate this, we apply a simple adaptive pooling kernel to compress the frame features into different token lengths, and train the model with varying numbers of frame tokens. The experimental results, as shown in Fig. 10, demonstrate minimal variation in performance across all sub-tasks in VNBench. This aligns with our initial design goal, which is to use VNBench primarily for evaluating the model's ability to understand temporal relationships in long contexts, rather than its spatial comprehension.

**Frame Sampling Number** Due to the constraints imposed by the current model's window size, most models rely on frame sampling to process video sequences. The frame sampling number determines how many raw video frames can be input into the model. We conducted an investigation into the effect of varying the number of sampled frames on the model's temporal understanding capabilities. As illustrated in Fig. 11, an increase in the number of sampled frames leads to performance improvements in the retrieval and ordering tasks on VNBench. This suggests that future research should focus on optimizing video models to process a greater number of frames and overcome current window limitations. On the other hand, the counting task exhibits only marginal improvements, indicating that while ncreasing the number of input frames enhances perceptual understanding, further progress in modeling higher-level and more complex temporal relationships (*e.g.*, spatio-temporal coherence) will require additional model optimization.

Table 3: Model analysis on temporal prompt. {idx} means the index of the sampled frame/image. HH:MM:SS means the timestamp of the sampled frames in the raw video.

| Prompt | RET | ORD | CNT | Overall |
|---|---|---|---|---|
| <image> | 62.4 | 12.2 | 13.3 | 29.3 |
| image {idx}: <image> | 64.0 | 18.7 | 19.3 | 34.0 |
| image {idx}: <image>
frame {idx}: <frame> | 65.5 | 19.9 | 18.4 | 34.7 |
| image: <image>
frame time HH:MM:SS: <frame> | 64.0 | 21.9 | 23.5 | 36.5 |

**Temporal Prompt** We also investigated the impact of temporal prompts on enhancing the temporal awareness of video models. Temporal prompts provide explicit textual cues that inform the language model about time-related aspects of the video, such as the order of frames or their corresponding timestamps. Our findings indicate that this straightforward approach significantly improves the model's performance in both ordering and counting tasks in Table 3. This suggests that the temporal modeling capabilities of video models can be effectively enhanced by incorporating prompts that inject time-related information.

## 7 LIMITATIONS

VideoNIAH offers both scalability and flexibility, enabling video model researchers to design increasingly complex construction rules tailored to their needs, facilitating more comprehensive evaluations of video models. We hope our work serves as a catalyst for the rapid iteration and optimization of video model capabilities. On the other hand, traditional comprehensive benchmarks still hold irreplaceable value. They are based on real-world videos with human-validated questions that reflect authentic scenarios, which synthetic data generated by the VideoNIAH framework cannot fully replicate. We believe that real-world benchmarks should complement synthetic ones created using the VideoNIAH framework, working together to advance research in video models.

## 8 CONCLUSION

In this paper, we propose a scalable synthetic framework for benchmarking video MLLMs, named VideoNIAH, which decouples the relationship between video content and query-response pairs. It also separates different aspects of video understanding skills, allowing us to probe the strengths and weaknesses of video MLLMs. VideoNIAH can evaluate various dimensions of video comprehension and can be applied to diverse video sources and lengths. Utilizing this framework, we construct the first synthetic video benchmark, VNBench, which assesses video model capabilities (temporal perception, chronological order, spatio-temporal conherence) across a broad range of video contexts. We provide a comprehensive evaluation of both proprietary and open-source video MLLMs, revealing that they still struggle with long-distance dependency tasks on VNBench. Additionally, we conduct model-level analysis and offer valuable insights for improving video MLLM training. We believe our work will inspire future advancements in the field.

## 9 ACKNOWLEDGMENT

This research is supported by Artificial Intelligence National Science and Technology Major Project(2023ZD0121200), the National Natural Science Foundation of China (No. 62437001, 62436001), the Key Research and Development Program of Jiangsu Province under Grant BE2023016-3, the Natural Science Foundation of Jiangsu Province under Grant BK20243051.

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

## Overview of Appendix

## A  DETAILS IN VNBENCH CONSTRUCTION

### A.1  SUBTITLE NEEDLE

The subtitles we used have a unified format:

The secret word is NAME.

while the candidate names are listed below.

> **Name Candidates**
>
> "Alice", "Bob", "Carol", "Dave", "Eve", "Frank", "Grace", "Harry", "Ivy", "Jack", "Kate", "Leo", "Mary", "Nick", "Olivia", "Paul", "Quentin", "Rachel", "Sam", "Tom", "Uma", "Victor", "Wendy", "Xander", "Yvonne", "Zach"

### A.2  IMAGE NEEDLE

The fruit images we used are shown below:

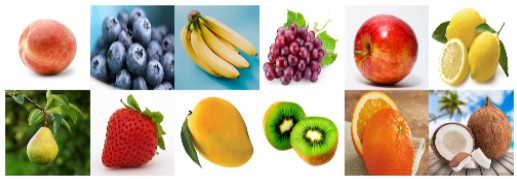

Figure 12: Fruit Image Candidates.

Furthermore, we have also gathered object images from the MSCOCO dataset and landmark images from Seed-Bench2.

### A.3  QUERY-RESPONSE GENERATION

For Retrieval tasks, negative answer options are randomly selected from the pool of needle candidates. For ordering tasks, the negative answer options consist of a shuffled version of the ground-truth answers. For counting tasks, we employ a sampling method based on a normal distribution centered around the ground-truth number, thereby increasing the difficulty of the multiple-choice problem.

Figure 13: Illusions of different tasks in VNBench. Each sample in VNBench consists of several inserted needles, query-response pairs generated by pre-defined rules, and a randomly selected video.

## A.4   CONSTRUCTION DETAILS

- **Retrieval-E**dit uses a human-made subtitle as the needle, which is similar to the method used in  (Reid et al., 2024). We sample a name word as the keyword and append it on the frames of a random video clip in the format "*The secret word is NAME*". The query in this task asks for the secret word stated in the inserted subtitle.

- **Retrieval-I**nsert uses an image as the needle. Unlike subtitles appended on video frames, these images are inserted between existing frames as static video clips. Furthermore, we divide this task into two levels according to image recognizability. The level-1 task uses common images, such as fruit images, as the image needle, while the level-2 task adopts more challenging images, *e.g.*, landmark images from SEED-Bench2 (Li et al., 2023a).

- **Ordering-E**dit uses human-made subtitles as the needles. We sample four different names used in the Retrieval-E task as the needles and ask the model to determine the correct order of these unique inserted names.

- **Ordering-I**nsert uses images as the needles. Four images are sampled to be inserted between existing frames as static video clips. Then, the model is required to give the correct temporal order of these image needles. The task is also divided into two levels according to image recognizability.

- **Counting-E**dit asks the model to count the appearing time of the object appended on the edited video segment. It is divided into 2 levels based on the task difficulty. The level-1 task uses human-made subtitles as the needles. We sample several names used in the Retrieval-E task as the needles and ask the model to provide the number of times the inserted subtitles appear. The level-2 task is the most challenging in VNBench. We choose one image from the candidate image

set and append it to four random video clips. In each video clip, this image can appear one to four times in different regions of the frame randomly. The model is asked to count the total number of appearances of this specified object, requiring counting in both spatial and temporal dimensions.

- **Counting-I**nsert uses images as the needles. We choose one image category and randomly sample several images from it as the image needles. We ask the model to provide the correct count of how many times one type of object appears in the video.

# B  TASK CORRELATION ANALYSIS

## B.1  SUB-TASK CORRELATION ANALYSIS

VNBench is constructed with the premise that different tasks can expose unique aspects in model performance. To affirm the legitimacy of these task categories and to facilitate the identification of key tasks, we conduct a task correlation analysis. Our evaluation encompasses ten distinct models, with each task being characterized by a vector that encapsulates the models' performance across a range of context sizes. These nine task vectors are subsequently organized through an agglomerative clustering technique, where the correlation coefficient is utilized as the measure of distance. As depicted in Fig. 15, the tasks within each of the two main categories (Retrieval and Ordering) tend to cluster together in a cohesive manner, devoid of overlap. On the other hand, the sub-tasks within the Counting category appear to be more autonomous, a phenomenon attributed to the distinct types of 'needles' employed in these tasks. The Counting-Edit-Level1 task leverages subtitles as needles to detect subtleties within individual frames, while the Counting-Insert task employs static frames as needles to gather information from within a single frame. Conversely, the Counting-Edit-Level2 task demands the insertion of multiple image needles across various frames, necessitating an advanced capacity for capturing both spatial and temporal details.

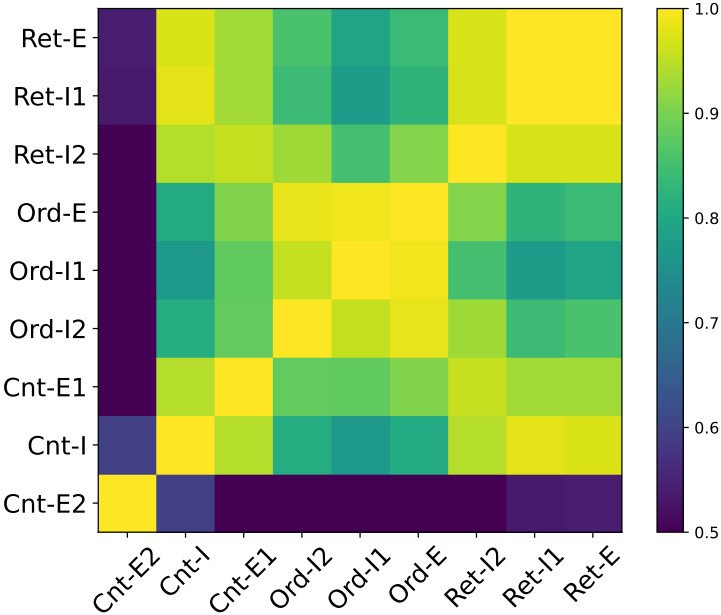

Figure 14: Sub-task correlation among 9 VNBench tasks.

## B.2 CROSS-TASK CORRELATION ANALYSIS

Here, we also analyzed the correlation between VNBench and other real-world video understanding benchmarks. Specifically, we selected six models with different numbers of sampled frames (Section 6), including models with 16, 32, 48, 64, 96, and 128 frames as input. These models were evaluated on real-world video understanding benchmarks, including VideoMME Fu et al. (2024), MLVU Zhou et al. (2024), and LongVideoBench Wu et al. (2024).

For each benchmark, we calculated the correlation coefficients of the corresponding performance vectors. The results show that all benchmarks achieved correlation coefficients above 0.75. Notably, VNBench demonstrated correlation coefficients higher than 0.9 with MLVU and VideoMME, two commonly used real-world benchmarks. This indicates that VNBench can, to a certain extent, reflect a model's ability to process real-world videos.

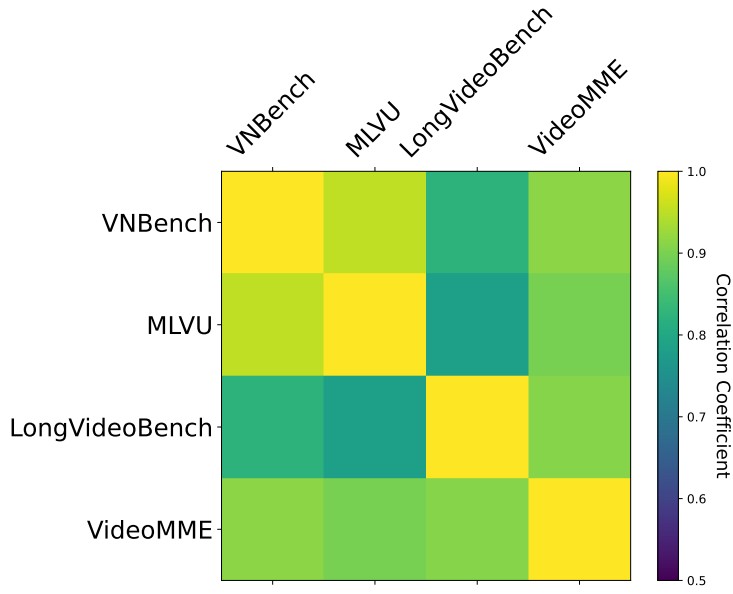

Figure 15: Cross-task correlation among VNBench and real-world video understanding tasks.

|  | VNBench | MLVU | LongVideoBench | VideoMME |
|---|---|---|---|---|
| VNBench | 1.000000 | 0.951426 | 0.823635 | 0.912259 |
| MLVU | 0.951426 | 1.000000 | 0.782742 | 0.897072 |
| LongVideoBench | 0.823635 | 0.782742 | 1.000000 | 0.908773 |
| VideoMME | 0.912259 | 0.897072 | 0.908773 | 1.000000 |

Table 4: Cross-task correlation matrix among VNBench and real-world video understanding tasks.

| #Frames | VNBench | MLVU | LongVideoBench | VideoMME |
|---|---|---|---|---|
| 16 | 23.33 | 52.53 | 46.78 | 49.74 |
| 32 | 29.93 | 54.84 | 47.16 | 49.96 |
| 48 | 34.15 | 56.22 | 48.75 | 52.81 |
| 64 | 32.59 | 57.23 | 47.08 | 52.59 |
| 96 | 37.26 | 60.97 | 48.60 | 53.26 |
| 128 | 39.70 | 61.44 | 51.40 | 56.11 |

Table 5: Evaluation results on VNBench and real-world benchmarks.

## C    EVALUATION DETAILS

### C.1    BASELINE MODELS

We evaluate 10 video MLLMs, including 7 open-source models and 3 proprietary models.

**Gemini 1.5 Pro** is a generative model capable of processing contexts up to 1 million tokens, accommodating videos as long as one hour. We employ a sampling strategy of 1 frame per second, and all frames are fed into the model along with their timestamps.

**GPT-4** is a multimodal generative model with a context window of approximately 128k tokens, capable of handling low-quality videos of about 5 minutes in length. We utilize a 1 frame per second sampling rate for GPT-4, processing all frames through the model. The version of GPT-4 we used including `gpt-4-turbo-2024-04-09` and `gpt-4o-2024-05-13`, represented as GPT-4o and GPT-4-turbo.

**VideoChatGPT** is a multimodal generative model, equipped with a context window of approximately 4k tokens. It employs the CLIP ViT-L/14 model for frame extraction, sampling 100 frames from the entire video.

**Video-LLaMA2** is built on top of BLIP-2 and MiniGPT-4. It is composed of two core components: Vision-Language (VL) Branch and Audio-Language (AL) Branch. In our research, we have exclusively utilized the VL branch, which processes 8 input frames.

**LLaMA-VID** incorporates a 4K context window. It employs EVA-CLIP-Giant to extract one context and one content token for each specified frame. It adopt a sampling rate of 1fps for the given video.

**Video-LLaVA** also employs a context window of 4k tokens. It leverage LanguageBind to extract the features of 8 uniformly sampled frames.

**VideoChat2** leverages a video encoder UMT-L to extract the features of uniformly sampled frames. We use 16 frame in our main evaluation experiment. We use position interpolation to fit our input frame number.

**ST-LLM** leverages BLIP-2 to extract the features of uniformly sampled frames. We use 32 frame in our main evaluation experiment.

**LLaVA-NeXT-Video** incorporates a context window comprising 4k tokens and employs CLIP ViT-L/14 to extract features from 32 evenly distributed frames. In our experiment, we sample video frames with 1fps sampling strategy. If the video contains more than 32 frames, we uniformly sample 32 frames from them. In our main evaluation experiment, we utilize the 7B model.

**Qwen2-VL** supports arbitrary image resolutions, mapping them into dynamic visual tokens for human-like visual processing. It uses Multimodal Rotary Position Embedding (M-ROPE) to handle positional information across text, images, and videos. We use 224×224 frame resolution and 1-fps frame sampling strategy for all video inputs.

**LLaVA-OneVision** is an open large multimodal model designed based on insights from the LLaVA-NeXT series. It pushes the performance boundaries in single-image, multi-image, and video scenarios simultaneously. The model excels at transfer learning across different modalities, showcasing strong video understanding and cross-scenario capabilities. We use a unified 64-frame sampling strategy for 0.5B, 7B and 72B models

### C.2    INFERENCE SETTING

For most video MLLMs, we use a unified prompt to get the answer for each sample:

> **Unified Inference Prompt Template**
>
> <QUESTION>
> A. <OPTION1>
> B. <OPTION2>

C. <OPTION3>

D. <OPTION4>

Answer with the option's letter from the given choices directly.

For most models, we use a rule-based matching strategy to extract option letter from their response. However, some models such as VideoChatGPT and Video-LLaMA2, can not follow the instructions to output letter. They are tend to output direct answers for the questions. Thus we use GPT-3.5 (version `gpt-3.5-turbo-0613`) as the judge to identify whether they can correctly answer the question.

**GPT Judge Prompt Template**

**SYSTEM:**

You are an intelligent chatbot designed for evaluating the correctness of generative outputs for question-answer pairs. Your task is to compare the predicted answer with the correct answer and determine if they match meaningfully. Here's how you can accomplish the task:

——

##INSTRUCTIONS:

- Focus on the meaningful match between the predicted answer and the correct answer.

- Consider synonyms or paraphrases as valid matches.

- Evaluate the correctness of the prediction compared to the answer.

**USER:**

Please evaluate the following video-based question-answer pair:

Question: <QUESTION>

Correct Answer: <GT ANSWER>

Predicted Answer: <PREDICTED ANSWER>

If the predicted answer expresses the same meaning as the correct answer, please output 1; otherwise, output 0.

DO NOT PROVIDE ANY OTHER OUTPUT TEXT OR EXPLANATION. Only provide 0 or 1.

## D    EVALUATION RESULTS ON VNBENCH-LONG

VNBench-Long is a synthetic benchmark contructed with VideoNIAH method. It contains 4 tasks in VNBench, including Retrieval-Edit, Retrieval-Insert, Ordering-Edit and Ordering-Insert. Unlike the primary test set, the video haystack in VNBench-Long ranges from 10 to 30 minutes, surpassing the context window limitations of most video MLLMs, except for Gemini 1.5 Pro, which can handle up to 1M tokens simultaneously. Therefore, we primarily report task performance on Gemini 1.5 Pro. Performance comparisons across different time intervals are shown in Fig. 16 . We observe that Gemini 1.5 Pro maintains stable performance even when video durations extend to 30 minutes, demonstrating robust long-context video understanding capabilities.

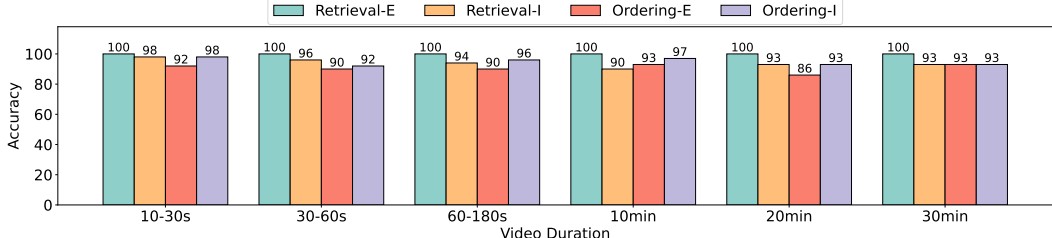

Figure 16: Evaluation Results on VNBench-Long

## E    EVALUATION RESULTS ON VNBENCH-ACT

VNBench-Act is a synthetic benchmark constructed using the VideoNIAH method. In VNBench-Act, we aim to evaluate the insertion of short video clips containing continuous natural frames from Something-Something V2 Goyal et al. (2017). This ensures that the inserted needle not only carries static image-level semantics but also conveys short-term action meanings.

It is worth noting that when the needles are extended to short video clips, the task becomes more challenging, especially for ordering tasks. This highlights limitations in modeling temporal action sequences effectively.

| Method | Ret-Action | Ord-Action | Cnt-Action |
|---|---|---|---|
| Video-LLaVA Lin et al. (2023) | 27.3 | 0.7 | 7.3 |
| LLaVA-NeXT-Video Zhang et al. (2024) | 42.7 | 0.7 | 11.3 |
| ST-LLM Liu et al. (2024d) | 54.7 | 0.0 | 22.7 |
| GPT-4o OpenAI (2023) | 85.6 | 11.3 | 10.0 |

Table 6: Evaluation Results on VNBench-Act.

## F EVALUATION ROBUSTNESS ANALYSIS

### F.1 CIRCULAR EVALUATION

In this analysis, we assess the robustness of the top two models in our test, Gemini 1.5 Pro and GPT-4o. We adjusted the number of iterations in our circular test from 1 to 4 to demonstrate the robustness of video MLLM inference, as shown in Table 7. We observed that increasing the number of iterations effectively reduces randomness in the MLLM's inference process, leading to a more accurate and fair evaluation.

Table 7: Evaluation robustness analysis on VNBench. We report the top-2 model on VNBench, including Gemini 1.5 Pro and GPT-4o. For each model, we report the task accuracy on 1 to 4 iteration number in circular evaluation.

| Video MLLMs | Retrieval | | | | Ordering | | | | Counting | | | | Overall |
|---|---|---|---|---|---|---|---|---|---|---|---|---|---|
| | E | I-1 | I-2 | Avg. | E | I-1 | I-2 | Avg. | E-1 | E-2 | I | Avg. | |
| *Proprietary Models* | | | | | | | | | | | | | |
| Gemini 1.5 Pro 4try | 100.0 | 96.0 | 76.0 | 90.7 | 90.7 | 95.3 | 32.7 | 72.9 | 60.7 | 7.3 | 42.0 | 36.7 | 66.7 |
| Gemini 1.5 Pro 3try | 100.0 | 98.0 | 78.0 | 92.0 | 94.0 | 95.3 | 44.6 | 78.0 | 64.7 | 10.6 | 46.0 | 40.4 | 70.1 |
| Gemini 1.5 Pro 2try | 100.0 | 98.0 | 80.0 | 92.6 | 96.7 | 96.0 | 60.7 | 84.4 | 71.3 | 16.7 | 51.3 | 46.4 | 74.5 |
| Gemini 1.5 Pro 1try | 100.0 | 98.0 | 87.3 | 95.1 | 98.0 | 96.7 | 72.0 | 88.9 | 80.7 | 24.7 | 58.0 | 54.4 | 79.5 |
| GPT-4o 4try | 100.0 | 98.0 | 87.3 | 95.3 | 88.4 | 86.6 | 45.2 | 73.4 | 36.8 | 0.0 | 36.1 | 24.5 | 64.4 |
| GPT-4o 3try | 100.0 | 98.7 | 87.3 | 95.3 | 91.2 | 90.6 | 53.3 | 78.3 | 44.2 | 2.7 | 38.8 | 28.6 | 67.4 |
| GPT-4o 2try | 100.0 | 98.7 | 88.7 | 95.7 | 92.5 | 92.7 | 62.8 | 82.7 | 50.3 | 11.4 | 47.5 | 36.4 | 71.6 |
| GPT-4o 1try | 100.0 | 98.3 | 92.0 | 97.1 | 95.2 | 96.0 | 76.3 | 89.2 | 61.7 | 28.9 | 55.6 | 48.7 | 78.3 |

### F.2 NEEDLE CONTENT

In VNBench, our goal is to decouple needle identification from the video understanding abilities we aim to evaluate (*e.g.*, temporal ordering). To this end, we use easy-to-identify objects as needles in several VNBench sub-tasks (*e.g.*, Retrieval-E, Retrieval-I1, Ordering-E, Ordering-I1, Counting-E1). To verify the robustness of these inserted needles, we replace the needle content and insert them into the same positions within video haystacks.

Specifically, for insertion tasks, we replace fruit images (as shown in Fig. 17) with animal images.

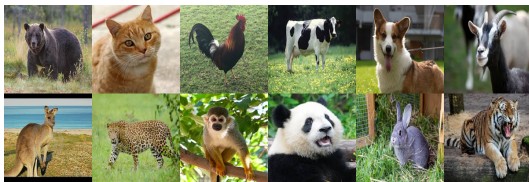

Figure 17: Animal Image Candidates.

Additionally, we replace subtitles (as shown in Appendix A.1) with a newly curated subtitle in editing tasks:

The private key is OBJECT.

where the object name is randomly sampled in candidates below:

> **Object Candidates**
>
> "apple", "banana", "cherry", "desert", "eagle", "forest", "garden", "harmony", "island", "jungle", "kite", "lemon", "mountain", "nectar", "ocean", "planet", "quartz", "river", "sunset", "tulip", "umbrella", "village", "waterfall", "xylophone", "yogurt", "zebra"

We compare the evaluation results across different types of needle content in Table 8. We calculated the correlation coefficients between the results before and after replacing the needle content. Experimental results indicate that the impact of needle content on test outcomes is minimal. This demonstrates the effectiveness of our decoupling strategy, ensuring that the evaluation focuses on the video understanding capabilities we aim to assess rather than being influenced by simple object recognition.

Table 8: Impact of needle content. We evaluated the impact of needle content on test results across five tasks: Retrieval-E, Retrieval-I1, Ordering-E, Ordering-I1, and Counting-E1. `src` refers to the original VNBench, and `new` refers to the test results after replacing the needle content. We computed task vectors for the five models' test results on these tasks and calculated the correlation coefficients between `src` and `new`.

| Video MLLMs | Retrieval-E | | Retrieval-I-1 | | Ordering-E | | Ordering-I2 | | Counting-E1 | |
|---|---|---|---|---|---|---|---|---|---|---|
| | src | new | src | new | src | new | src | new | src | new |
| Gemini 1.5 Pro | 100.0 | 100.0 | 96.0 | 98.0 | 90.7 | 92.0 | 32.7 | 31.3 | 60.7 | 59.3 |
| GPT-4o | 100.0 | 100.0 | 98.0 | 96.0 | 88.4 | 90.7 | 45.2 | 46.0 | 36.8 | 36.0 |
| LLaVA-NeXT-Video-7B | 56.7 | 55.3 | 56.7 | 60.0 | 0.7 | 0.7 | 0.7 | 0.7 | 6.7 | 3.3 |
| ST-LLM | 58.0 | 59.3 | 64.7 | 66.7 | 0.0 | 0.0 | 0.0 | 0.0 | 21.3 | 20.6 |
| Video-LLaVA-7B | 26.0 | 22.0 | 28.0 | 26.0 | 0.7 | 0.7 | 2.0 | 2.0 | 16.7 | 15.3 |
| **Correlation Coefficient** | 0.9990 | | 0.9965 | | 0.9999 | | 0.9993 | | 0.9990 | |

## F.3 SAMPLE NUMBER

In this section, we investigate the effect of sample size on VNBench. The original VNBench consists of 1350 samples. To evaluate the impact of a larger dataset, we curated a new benchmark using the same methodology, which includes 2700 samples in total. We tested five models on the expanded benchmark and compared the results with those from the original split, as shown in Table 9. Our analysis reveals a relatively high correlation coefficient between the source dataset and the enlarged benchmark, indicating the robustness of our benchmark with respect to the current sample size.

Table 9: Impact of sample number. We evaluated the impact of sample size on the average accuracy across three VNBench tasks: Retrieval, Ordering, and Counting. `src` refers to the original VNBench, which consist of 1350 samples, while `more` refers to the newly curated benchmark, which follows the same methodology but includes 2700 samples. We computed task vectors for the test results of five models across these tasks and calculated the correlation coefficients between `src` and `more`.

| Video MLLMs | Retrieval | | Ordering | | Counting | | Overall | |
|---|---|---|---|---|---|---|---|---|
| | src | more | src | more | src | more | src | more |
| Gemini 1.5 Pro | 90.7 | 89.2 | 72.9 | 73.2 | 36.7 | 35.7 | 66.7 | 66.0 |
| GPT-4o | 95.3 | 96.7 | 73.4 | 75.2 | 24.5 | 24.3 | 64.4 | 65.4 |
| LLaVA-NeXT-Video-7B | 44.2 | 42.4 | 0.4 | 2.3 | 15.5 | 13.5 | 20.1 | 19.4 |
| ST-LLM | 51.3 | 49.4 | 0.0 | 0.5 | 16.7 | 15.7 | 22.7 | 21.9 |
| Video-LLaVA-7B | 23.8 | 23.8 | 1.1 | 2.6 | 12.4 | 11.1 | 12.4 | 12.5 |
| **Correlation Coefficient** | 0.9998 | | 0.9577 | | 0.9990 | | 0.9996 | |

## G TRAINING SETTING IN MODEL ANALYSIS

For our trained MLLM, we used siglip-so400m-patch14-384 (Zhai et al., 2023) as the visual encoder for frames, with mean pooling and an MLP as the connection layer. The visual features from different frames were concatenated and combined with textual instructions before being input into the LLM, which utilized Qwen2-7B (Yang et al., 2024).

We sample the input videos into several frames and encode each frame into a fixed length of visual features with the visual encoder. An MLP modality projector then maps these visual features to visual tokens.

$$\boldsymbol{X}_V = \left[\text{MLP}(\text{Pooling}(\boldsymbol{F}_1^{\text{Frame}})), \ldots, \text{MLP}(\text{Pooling}(\boldsymbol{F}_N^{\text{Frame}}))\right] \tag{1}$$

These visual tokens, along with human-provided textual instructions, are fed into a large language model (LLM) to perform advanced video understanding tasks.

We train the model with visual instructions and responses to better understand human instructions in visual contexts. The loss function is defined as follows:

$$\text{Loss} = -\sum_{\substack{i=1 \\ i \in \boldsymbol{X}_{\text{ans}}}}^{L} \log P_\theta \left(x_i \mid \boldsymbol{X}_V, \boldsymbol{X}_{\text{ins},<i}, \boldsymbol{X}_{\text{ans},<i}\right) \tag{2}$$

where $L$ is the sequence length, $\boldsymbol{X}_{\text{ans},<i}$ and $\boldsymbol{X}_{\text{ins},<i}$ represent the tokens from the answer and instruction sequences preceding the $i$-th token, and $\theta$ denotes the entire set of model parameters.

We trained the video MLLM using the dataset shown in Table 10 with 8192 token context length. The training hyperparameters included a global batch size of 64, with the learning rates set to 2e-5 for the LLM, 1e-4 for the MLP, and 2e-6 for the visual encoder. A cosine learning rate schedule was applied. All models are trained on 16 NVIDIA A100 GPUS with 1 epoch.

| Modality | Dataset | Samples |
|---|---|---|
| Image-Text | Cauldron | 1.8M |
| Video-Text | VideoChatGPT-100K | 100K |
| | ShareGPT4Video | 40K |
| | ShareGPTVideo | 255K |
| | VIM | 32K |
| | NExT-QA | 40K |
| | SthSthV2 | 40K |
| | STAR | 40K |
| | TextVR | 40K |
| | CLEVRER | 80K |
| | Kinetics-710 | 40K |
| Total | - | 2.5M |

Table 10: The statistics of our training data, including 1.8M image-text instructions and 0.7M video-text instructions.

# H    COMPLETE RESULTS ON VNBENCH

Table 11: Complete evaluation results on VNBench. VNBench includes 3 synthetic tasks constructed with VideoNIAH method, while each task is divided into 3 splits. We evaluate 3 proprietary models and 9 open-source models in total.

| Video MLLMs | Retrieval | | | | Ordering | | | | Counting | | | | Overall |
|---|---|---|---|---|---|---|---|---|---|---|---|---|---|
| | E | I-1 | I-2 | Avg. | E | I-1 | I-2 | Avg. | E-1 | E-2 | I | Avg. | |
| *Video Haystack Length: 10-30s* | | | | | | | | | | | | | |
| Gemini 1.5 Pro | 100.0 | 98.0 | 80.0 | 92.7 | 92.0 | 98.0 | 32.0 | 74.0 | 66.0 | 4.0 | 46.0 | 38.7 | 68.4 |
| GPT-4o | 100.0 | 98.0 | 90.0 | 96.0 | 100.0 | 93.9 | 57.1 | 83.7 | 46.0 | 0.0 | 48.0 | 31.3 | 70.3 |
| GPT-4-turbo | 100.0 | 100.0 | 86.0 | 95.3 | 38.0 | 20.4 | 26.5 | 28.3 | 34.0 | 0.0 | 40.0 | 24.7 | 49.4 |
| LLaMA-VID | 8.0 | 18.0 | 28.0 | 18.0 | 0.0 | 0.0 | 0.0 | 0.0 | 18.0 | 2.0 | 14.0 | 11.3 | 9.8 |
| Video-LLaVA | 40.0 | 40.0 | 30.0 | 36.7 | 0.0 | 2.0 | 6.0 | 2.7 | 26.0 | 2.0 | 28.0 | 18.7 | 19.3 |
| VideoChat2 | 82.0 | 76.0 | 26.0 | 61.3 | 0.0 | 0.0 | 4.0 | 1.3 | 2.0 | 2.0 | 6.0 | 3.3 | 22.0 |
| LLaVA-NeXT-Video-7B | 90.0 | 74.0 | 36.0 | 66.7 | 0.0 | 0.0 | 0.0 | 0.0 | 12.0 | 16.0 | 30.0 | 19.3 | 28.7 |
| ST-LLM | 88.0 | 76.0 | 48.0 | 70.7 | 0.0 | 0.0 | 0.0 | 0.0 | 28.0 | 0.0 | 36.0 | 21.3 | 30.7 |
| Qwen2-VL-7B | 100 | 78.0 | 44.0 | 74.0 | 6.0 | 12.0 | 4.0 | 7.3 | 20.0 | 8.0 | 28.0 | 18.7 | 33.3 |
| LLaVA-OneVision-0.5B | 100.0 | 96.0 | 22.0 | 72.7 | 6.0 | 4.0 | 4.0 | 4.7 | 6.0 | 10.0 | 22.0 | 12.7 | 30.0 |
| LLaVA-OneVision-7B | 100.0 | 100.0 | 72.0 | 90.7 | 94.0 | 78.0 | 58.0 | 76.7 | 64.0 | 12.0 | 32.0 | 36.0 | 67.8 |
| LLaVA-OneVision-72B | 100.0 | 98.0 | 76.0 | 91.3 | 96.0 | 98.0 | 66.0 | 86.7 | 52.0 | 12.0 | 36.0 | 33.3 | 70.4 |
| *Video Haystack Length: 30-60s* | | | | | | | | | | | | | |
| Gemini 1.5 Pro | 100.0 | 96.0 | 80.0 | 92.0 | 90.0 | 92.0 | 32.0 | 71.3 | 60.0 | 8.0 | 42.0 | 36.7 | 66.7 |
| GPT-4o | 100.0 | 100.0 | 88.0 | 96.0 | 91.8 | 86.0 | 52.0 | 76.6 | 42.0 | 0.0 | 40.0 | 27.3 | 66.6 |
| GPT-4-turbo | 100.0 | 100.0 | 80.0 | 93.3 | 55.1 | 18.0 | 18.0 | 30.4 | 40.0 | 0.0 | 38.0 | 26.0 | 49.9 |
| LLaMA-VID | 4.0 | 14.0 | 16.0 | 11.3 | 0.0 | 0.0 | 0.0 | 0.0 | 10.0 | 0.0 | 14.0 | 8.0 | 6.4 |
| Video-LLaVA | 18.0 | 16.0 | 8.0 | 14.0 | 0.0 | 0.0 | 0.0 | 0.0 | 16.0 | 0.0 | 22.0 | 12.7 | 8.9 |
| VideoChat2 | 38.0 | 34.0 | 14.0 | 28.7 | 0.0 | 0.0 | 0.0 | 0.0 | 6.0 | 0.0 | 10.0 | 5.3 | 11.3 |
| LLaVA-NeXT-Video | 54.0 | 60.0 | 12.0 | 42.0 | 0.0 | 0.0 | 0.0 | 0.0 | 8.0 | 8.0 | 34.0 | 16.7 | 19.6 |
| ST-LLM | 60.0 | 80.0 | 30.0 | 56.7 | 0.0 | 0.0 | 0.0 | 0.0 | 22.0 | 0.0 | 32.0 | 18.0 | 24.9 |
| Qwen2-VL-7B | 98.0 | 70.0 | 30.0 | 66.0 | 18.0 | 14.0 | 14.0 | 15.3 | 24.0 | 14.0 | 30.0 | 22.7 | 34.7 |
| LLaVA-OneVision-0.5B | 100.0 | 90.0 | 26.0 | 72.0 | 0.0 | 0.0 | 4.0 | 1.3 | 12.0 | 4.0 | 22.0 | 12.7 | 28.7 |
| LLaVA-OneVision-7B | 100.0 | 100.0 | 56.0 | 85.3 | 92.0 | 58.0 | 42.0 | 64.0 | 46.0 | 10.0 | 36.0 | 30.7 | 60.0 |
| LLaVA-OneVision-72B | 100.0 | 100.0 | 56.0 | 85.3 | 96.0 | 94.0 | 70.0 | 86.7 | 52.0 | 22.0 | 38.0 | 37.3 | 69.8 |
| *Video Haystack Length: 60-180s* | | | | | | | | | | | | | |
| Gemini 1.5 Pro | 100.0 | 94.0 | 68.0 | 87.3 | 90.0 | 96.0 | 34.0 | 73.3 | 56.0 | 10.0 | 38.0 | 34.7 | 65.1 |
| GPT-4o | 100.0 | 98.0 | 84.0 | 94.0 | 73.5 | 80.0 | 26.5 | 60.0 | 22.3 | 2.0 | 20.4 | 14.9 | 56.3 |
| GPT-4-turbo | 100.0 | 98.0 | 80.0 | 92.7 | 34.7 | 30.0 | 24.5 | 29.7 | 38.8 | 0.0 | 18.4 | 19.1 | 47.2 |
| LLaMA-VID | 14.0 | 16.0 | 14.0 | 14.7 | 0.0 | 0.0 | 2.0 | 0.7 | 4.0 | 0.0 | 8.0 | 4.0 | 6.4 |
| Video-LLaVA | 20.0 | 28.0 | 14.0 | 20.7 | 2.0 | 0.0 | 0.0 | 0.7 | 8.0 | 0.0 | 10.0 | 6.0 | 9.1 |
| VideoChat2 | 10.0 | 10.0 | 4.0 | 8.0 | 0.0 | 0.0 | 0.0 | 0.0 | 2.0 | 0.0 | 8.0 | 3.3 | 3.8 |
| LLaVA-NeXT-Video | 26.0 | 36.0 | 10.0 | 24.0 | 2.0 | 0.0 | 2.0 | 1.3 | 0.0 | 20.0 | 12.0 | 10.7 | 12.0 |
| ST-LLM | 26.0 | 38.0 | 16.0 | 26.7 | 0.0 | 0.0 | 0.0 | 0.0 | 14.0 | 4.0 | 14.0 | 10.7 | 12.4 |
| Qwen2-VL-7B | 96.0 | 80.0 | 26.0 | 67.3 | 24.0 | 12.0 | 8.0 | 14.7 | 34.0 | 6.0 | 16.0 | 18.7 | 33.6 |
| LLaVA-OneVision-0.5B | 66.0 | 56.0 | 18.0 | 46.7 | 2.0 | 0.0 | 0.0 | 0.7 | 2.0 | 12.0 | 14.0 | 9.3 | 18.9 |
| LLaVA-OneVision-7B | 66.0 | 62.0 | 38.0 | 55.3 | 24.0 | 14.0 | 12.0 | 16.7 | 14.0 | 4.0 | 14.0 | 10.7 | 27.6 |
| LLaVA-OneVision-72B | 72.0 | 62.0 | 40.0 | 58.0 | 42.0 | 30.0 | 26.0 | 32.7 | 24.0 | 10.0 | 18.0 | 17.3 | 36.0 |
| *Video Haystack Length: 10 minutes* | | | | | | | | | | | | | |
| Gemini 1.5 Pro | 100.0 | 90.0 | - | - | 93.3 | 96.7 | - | - | - | - | - | - | - |
| *Video Haystack Length: 20 minutes* | | | | | | | | | | | | | |
| Gemini 1.5 Pro | 100.0 | 93.3 | - | - | 86.7 | 93.3 | - | - | - | - | - | - | - |
| *Video Haystack Length: 30 minutes* | | | | | | | | | | | | | |
| Gemini 1.5 Pro | 100.0 | 93.3 | - | - | 93.3 | 93.3 | - | - | - | - | - | - | - |

# I  TASK SAMPLES OF VNBENCH

The whole VNBench contain 9 tasks. Here we show samples of generated task data.

**Retrieval-Edit Task Sample**

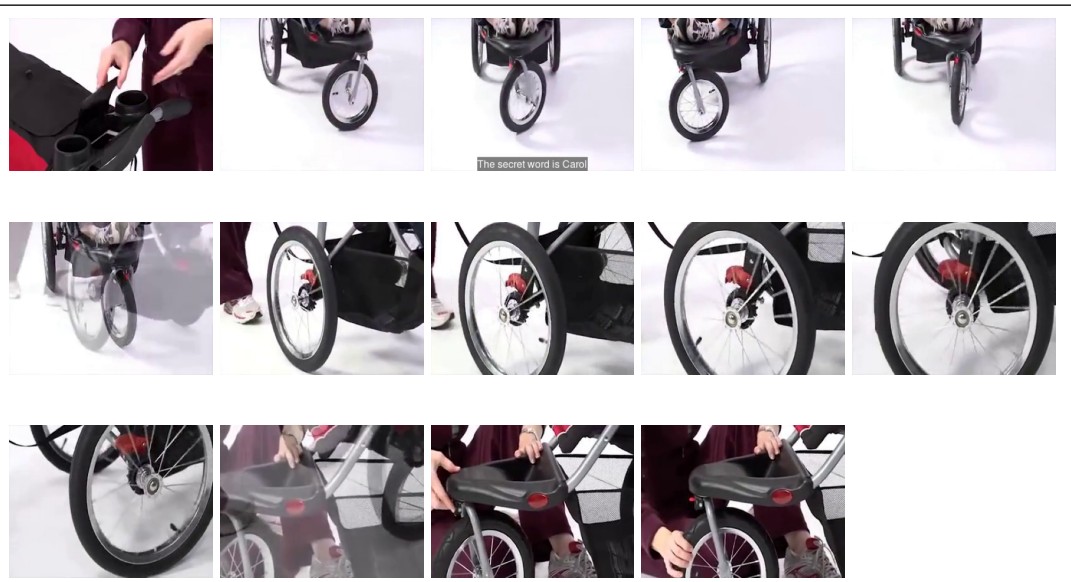

| Question: | What is the secret word in this video? |
|---|---|
| Options: | A. Rachel, B. Carol, C. Mary, D. Nick |
| Answer: | B. Carol. |

**Retrieval-Insert-Level1 Task Sample**

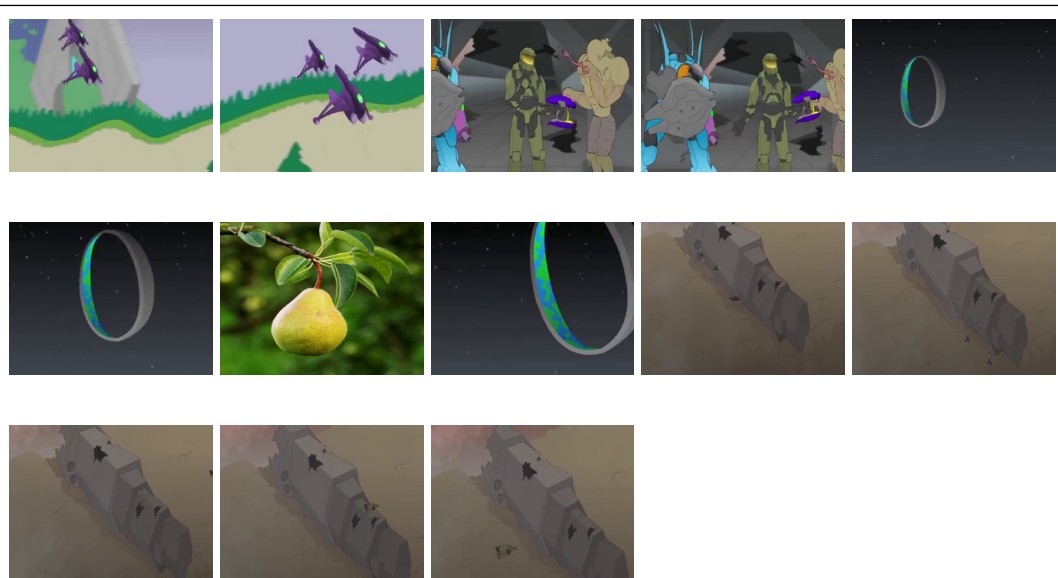

| Question: | What is the fruit that appears in this video? |
|---|---|
| Options: | A. Apple, B. Pear, C. Peach, D. Lemon |
| Answer: | B. Pear. |

**Retrieval-Insert-Level2 Task Sample**

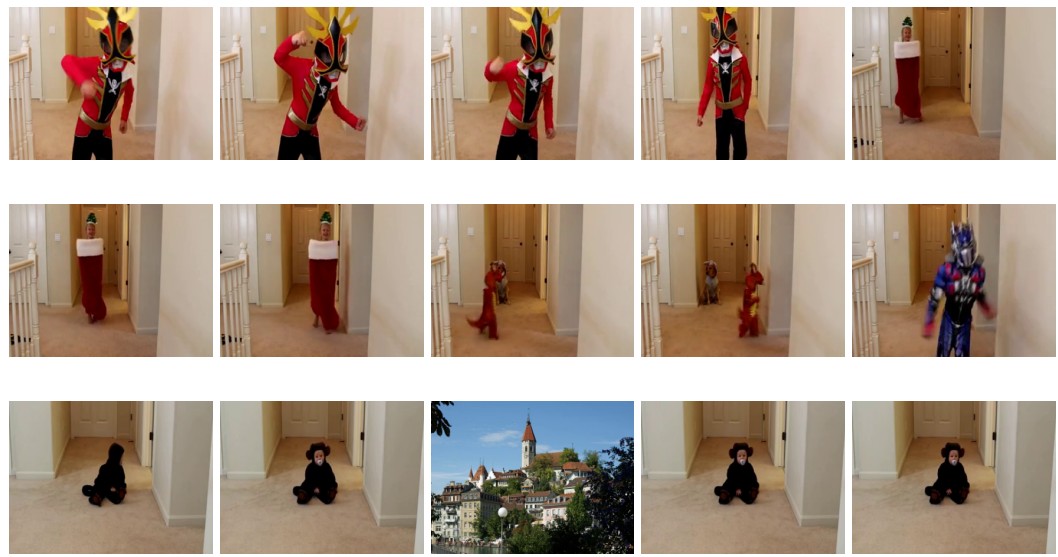

| Question: | What is the landmark that appears in this video? |
| Options: | A. Kagoshima Prefectural Kamoike Athletic Stadium |
| | B. Stadtkirche (Thun) |
| | C. Waterkant, Paramaribo |
| | D. Templo Khadro Ling (Budismo Tibetano), Treas Coroas, Brasil |
| Answer: | Stadtkirche (Thun). |

**Ordering-Edit Task Sample**

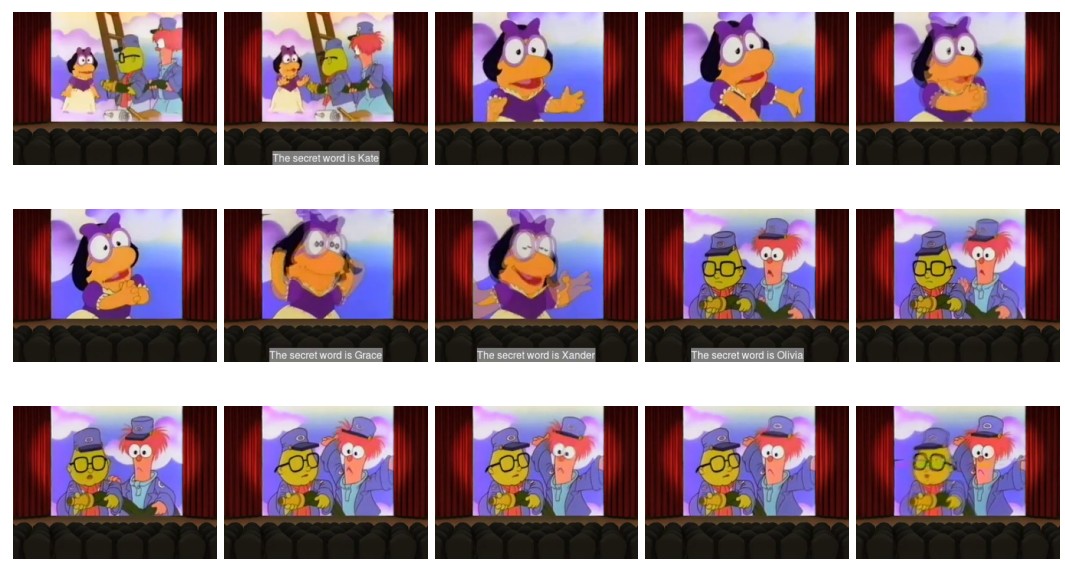

| Question: | What is the order of the secret words that appeared in the video? |
| Options: | A. Xander, Kate, Grace, Olivia |
| | B. Kate, Xander, Olivia, Grace |
| | C. Olivia, Xander, Kate, Grace |
| | D. Kate, Grace, Xander, Olivia |
| Answer: | D. Kate, Grace, Xander, Olivia. |

**Ordering-Insert-Level1 Task Sample**

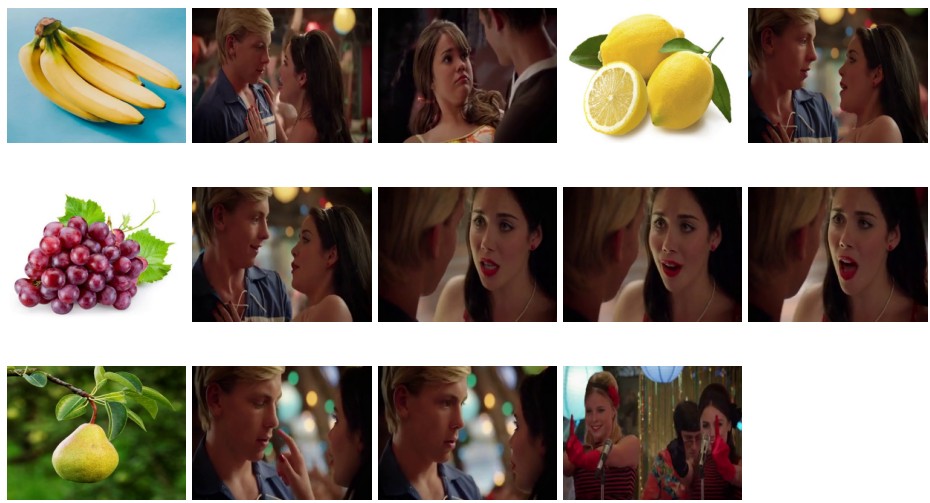

Question:  What is the order of fruits appearing in the video?
Options:   A. banana, grapes, lemon, pear
           B. banana, lemon, grapes, pear
           C. grapes, lemon, banana, pear
           D. banana, pear, lemon, grapes
Answer:    B. banana, lemon, grapes, pear.

**Ordering-Insert-Level2 Task Sample**

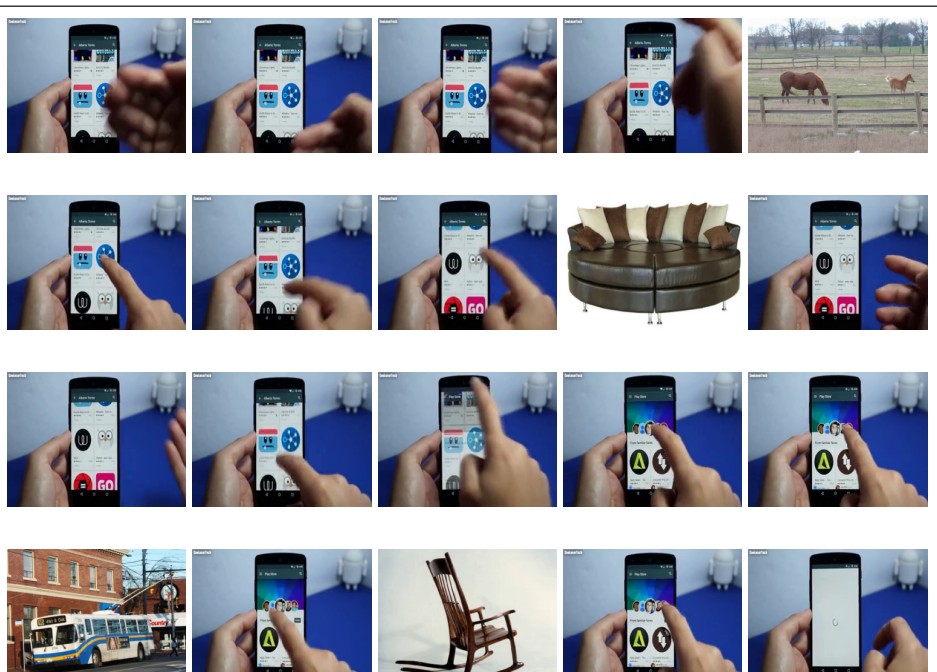

Question:  What is the order of images appearing in the video?
Options:   A. chair, sofa, horse, bus
           B. chair, sofa, bus, horse
           C. chair, bus, sofa, horse
           D. horse, sofa, bus, chair
Answer:    D. horse, sofa, bus, chair.

**Counting-Edit-Level1 Task Sample**

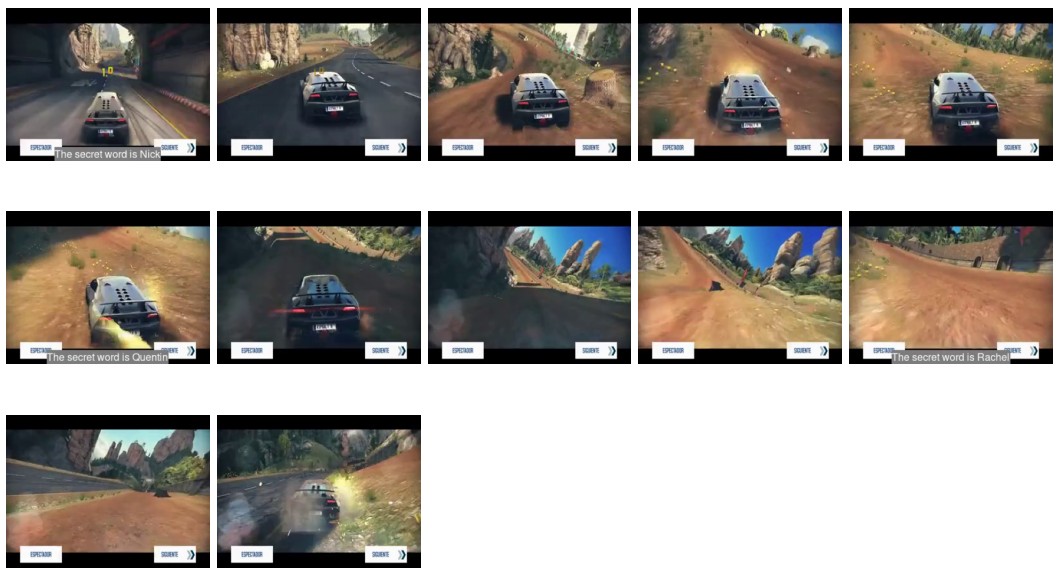

Question:  How many secret words appeared in the video?
Options:   A. 3, B. 6, C. 1, D. 5
Answer:    A. 3

**Counting-Edit-Level2 Task Sample**

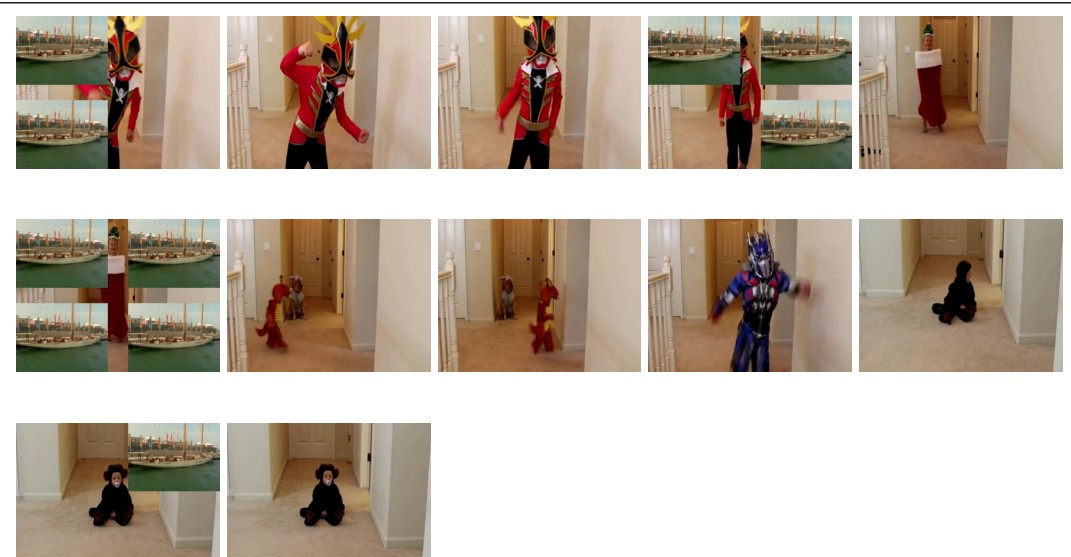

Question:  Some boats were inserted into 4 small sections of the video. How many boats appeared in the video in total?
Options:   A. 9, B. 11, C. 10, D. 12
Answer:    C. 10

**Counting-Insert Task Sample**

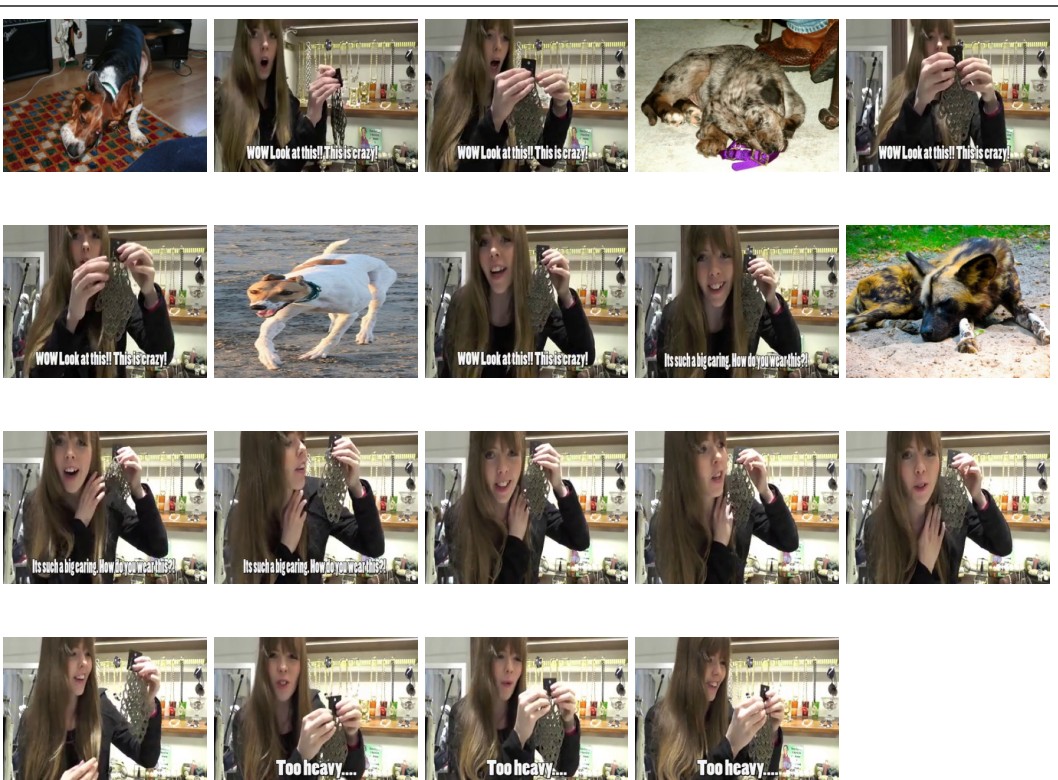

Question:    How many dogs appeared in the video?
Options:     A. 5, B. 4, C. 3, D. 6
Answer:      B. 4

## J    CONSISTENCY EVALUATION

Since our tasks are in a multi-choice QA format, and most video MLLMs (excluding VideoChat-GPT Maaz et al. (2023) and VideoLLaMA2 Zhang et al. (2023), which directly output sentence and must use GPT to evaluate) have strong instruction-following capabilities, their output is constrained to the four provided option letters. This means that even if the model cannot select the correct answer, it will still output one of the option letters.

As mentioned in Wang et al. (2024a); Wei et al. (2024), LLMs inherently have preferences for certain options, and different models show varying sensitivity to the order of options. When options are shuffled (in our circular evaluation) and a model lacks the ability to analyze the question but is forced to output an answer, it tends to rely on internal biases to select its preferred option letter rather than making a choice based on option content. This can lead to situations where the model outputs the same option for a shuffled version of the question, even though the content corresponding to that option differs.

We analyzed this phenomenon on 9 sub-tasks in Tables 12 to 14. First, we examined the option consistency of LLaVA-OneVision series, Gemini 1.5 Pro, and GPT-4 series. Since our circular evaluation involves four iterations with shuffled option orders, only predictions where the option content remains the same across all four iterations are considered consistent. We define **Consist.** as the proportion of consistent predictions across these four iterations, and **Acc.** as the accuracy of the model's predictions under circular evaluation.

In this setup, two outcomes are possible:

1. The predictions are consistent, meaning the model may predict the same answer as the GroundTruth (correct) or a different answer (incorrect).

2. The predictions are inconsistent, which results in the answer being marked as incorrect in circular evaluation.

Thus, accurate predictions are a subset of consistent predictions. We compute the conditional probability **Acc.|Consist.**, which represents the accuracy of the model's prediction given that the results are consistent.

Our conclusions are as follows:

1. When the conditional probability **Acc.|Consist.** is significantly higher than 25% (roughly equivalent to random guessing), the accuracy can effectively reflect the model's ability to handle the task. In Counting-E1 and Counting-I, the results from the model performance comparison are valuable.

2. When the conditional probability **Acc.|Consist.** fluctuates around 25% or falls below 25%, comparisons of accuracy have limited significance. In this case, the conclusion is that the models are unable to solve the task, meaning that all models perform unsatisfactorily on Counting-E2. This represents a common issue across all models and indicates an area where VideoMLLMs need optimization.

3. For GPT-series models, we find that compared to Gemini 1.5 Pro and LLaVA-OneVision series models, their **Consistency Rate** on unsolvable tasks (e.g., Counting-E2) is very low (5.3% for GPT-4o, 3.6% for GPT-4-turbo). This suggests that these models are highly sensitive to the order of options, resulting in a 0.0% accuracy in circular evaluation. While other models also fail to provide correct answers (reflected by the conditional probability **Acc.|Consist.** around 25%, roughly equivalent to random guessing), their higher **Consistency Rate** leads to slightly higher **Acc.** values. However, as concluded in point 2, such comparisons are not significant. This phenomenon can also be validated in Table 7, where GPT-4o shows performance closer to Gemini 1.5 Pro with fewer evaluation iterations.

4. The low **Consistency** performance of GPT-series models is seen only in tasks they cannot solve. On tasks they can handle, their **Consistency Rate** is high (in Table 12 and Table 13).

The above conclusions suggest that for VNBench, comparisons at high **Acc.** levels reflect the models' abilities, while low **Acc.** levels indicate a common issue for all Video MLLMs. In fact, this conclusion

holds true for most multi-choice QA benchmarks. Meanwhile, we believe that **Acc.|Consist.** may also be an important metric in circular evaluation.

Table 12: Accuracy and consistant rate of VideoMLLMs in multi-try circular evaluations of VNBench Retrieval Tasks.

| Video MLLMs | Retrieval-E1 | | | Retrieval-I1 | | | Retrieval-I2 | | |
|---|---|---|---|---|---|---|---|---|---|
| | Acc. | Consist. | Acc.\|Consist. | Acc. | Consist. | Acc.\|Consist. | Acc. | Consist. | Acc.\|Consist. |
| random choice | - | - | 25.0 | - | - | 25.0 | - | - | 25.0 |
| Gemini 1.5 Pro (Reid et al., 2024) | 100.0 | 100.0 | 100.0 | 96.0 | 96.7 | 99.3 | 76.0 | 80.0 | 95.0 |
| GPT-4o (OpenAI, 2023) | 100.0 | 100.0 | 100.0 | 98.0 | 100.0 | 98.0 | 87.3 | 88.0 | 99.2 |
| GPT-4-turbo (OpenAI, 2023) | 100.0 | 100.0 | 100.0 | 99.3 | 99.3 | 100.0 | 82.0 | 84.0 | 97.6 |
| LLaVA-OneVision-0.5B (Li et al., 2024) | 88.7 | 91.3 | 97.2 | 80.7 | 88.7 | 91.0 | 22.0 | 50.7 | 43.4 |
| LLaVA-OneVision-7B (Li et al., 2024) | 88.7 | 92.0 | 96.4 | 87.3 | 94.7 | 92.2 | 55.3 | 72.0 | 76.8 |
| LLaVA-OneVision-72B (Li et al., 2024) | 90.7 | 95.3 | 95.2 | 86.7 | 96.7 | 89.7 | 57.3 | 64.7 | 88.6 |

Table 13: Accuracy and consistant rate of VideoMLLMs in multi-try circular evaluations of VNBench Ordering Tasks.

| Video MLLMs | Ordering-E | | | Ordering-I1 | | | Ordering-I2 | | |
|---|---|---|---|---|---|---|---|---|---|
| | Acc. | Consist. | Acc.\|Consist. | Acc. | Consist. | Acc.\|Consist. | Acc. | Consist. | Acc.\|Consist. |
| random choice | - | - | 25.0 | - | - | 25.0 | - | - | 25.0 |
| Gemini 1.5 Pro (Reid et al., 2024) | 90.7 | 90.7 | 100.0 | 95.3 | 95.3 | 100.0 | 32.7 | 33.3 | 98.2 |
| GPT-4o (OpenAI, 2023) | 88.4 | 88.4 | 100.0 | 86.6 | 87.2 | 99.3 | 45.2 | 46.6 | 97.0 |
| GPT-4-turbo (OpenAI, 2023) | 42.6 | 43.2 | 98.6 | 22.8 | 24.8 | 91.9 | 23.0 | 24.3 | 95.8 |
| LLaVA-OneVision-0.5B (Li et al., 2024) | 2.7 | 5.3 | 50.9 | 1.3 | 3.3 | 39.4 | 2.7 | 5.3 | 50.9 |
| LLaVA-OneVision-7B (Li et al., 2024) | 70.0 | 73.3 | 96.4 | 50.0 | 53.3 | 93.8 | 37.3 | 40.7 | 91.6 |
| LLaVA-OneVision-72B (Li et al., 2024) | 78.0 | 80.7 | 96.7 | 74.0 | 76.0 | 97.4 | 54.0 | 56.7 | 95.2 |

Table 14: Accuracy and consistant rate of VideoMLLMs in multi-try circular evaluations of VNBench Counting Tasks.

| Video MLLMs | Counting-E1 | | | Counting-E2 | | | Counting-I | | |
|---|---|---|---|---|---|---|---|---|---|
| | Acc. | Consist. | Acc.\|Consist. | Acc. | Consist. | Acc.\|Consist. | Acc. | Consist. | Acc.\|Consist. |
| random choice | - | - | 25.0 | - | - | 25.0 | - | - | 25.0 |
| Gemini 1.5 Pro (Reid et al., 2024) | 60.7 | 67.3 | 90.2 | 7.3 | 27.3 | 26.7 | 42.0 | 54.0 | 77.8 |
| GPT-4o (OpenAI, 2023) | 36.8 | 42.9 | 85.8 | 0.0 | 5.3 | 0.0 | 36.1 | 55.0 | 72.2 |
| GPT-4-turbo (OpenAI, 2023) | 37.6 | 44.9 | 83.7 | 0.0 | 3.6 | 0.0 | 32.4 | 36.2 | 89.5 |
| LLaVA-OneVision-0.5B (Li et al., 2024) | 6.7 | 33.3 | 20.1 | 9.3 | 26.0 | 35.7 | 18.7 | 73.3 | 25.5 |
| LLaVA-OneVision-7B (Li et al., 2024) | 41.3 | 72.0 | 57.3 | 8.7 | 48.7 | 17.9 | 27.3 | 92.0 | 29.6 |
| LLaVA-OneVision-72B (Li et al., 2024) | 42.7 | 60.7 | 70.3 | 14.7 | 60.7 | 24.2 | 30.7 | 80.0 | 38.4 |

