# OpenReview forum: "Needle In A Video Haystack: A Scalable  Synthetic Evaluator for Video MLLMs"
_ICLR.cc/2025/Conference — ICLR 2025 Poster_

### Official Review · Reviewer_9VkG · 2024-11-03

**Soundness:** 2
**Presentation:** 2
**Contribution:** 3
**Rating:** 5
**Confidence:** 4

**Summary:**

This paper introduces VideoNIAH, a novel framework for constructing scalable synthetic benchmarks for evaluating video multimodal large language models (MLLMs).  The core idea is to inject unrelated visual or textual "needles" into existing videos ("haystacks") and then pose questions designed to test specific video understanding skills, such as temporal perception, chronological ordering, and spatio-temporal coherence.  The authors create VNBench, a benchmark based on VideoNIAH, and evaluate 12 proprietary and open-source video MLLMs, revealing significant performance gaps, particularly in long-dependency tasks.  They also conduct a detailed analysis of model performance across various factors like video length, needle number, and position.

**Strengths:**

1. In VideoNIAH, the decoupling of video content from query-response pairs addresses a limitation of existing real-world video benchmarks, namely the high cost of data creation and the difficulty in isolating specific skills.  The synthetic approach mitigates data leakage concerns. The framework's design allows for easy generation of diverse and large-scale benchmarks, adaptable to various video lengths and complexities.
2. The authors evaluate a wide range of models (both proprietary and open-source), providing a comprehensive comparison of their performance on different tasks and under varying conditions. The analysis goes beyond simple performance metrics, exploring the impact of various factors (video length, needle number, position, etc.) on model accuracy.  This provides valuable insights into the strengths and weaknesses of current video MLLMs.

**Weaknesses:**

1. While the synthetic nature is a strength in terms of scalability and avoiding data leakage, it also limits the ecological validity of the benchmark.  Performance on VNBench doesn't guarantee real-world performance. The authors acknowledge this, but a correlation computation between the given benchmark and other real-world ones is needed.
2. The reliance on CLIP for filtering needle images introduces a potential bias.  The performance of the CLIP model itself could influence the results, especially on synthetic videos.  A sensitivity analysis or exploration of alternative filtering methods would be valuable.
3. Minor issues
- Wrong single quotation marks in L18, P1.

**Questions:**

1. How does the performance on VNBench correlate with performance on established real-world video understanding benchmarks?  Can you provide a comparative analysis?
2. Have you explored the sensitivity of the results to the CLIP similarity threshold used for needle filtering?
3. How robust are the results to variations in the types and complexities of the "needles"?
4. Could the VideoNIAH framework be adapted to evaluate other multimodal tasks beyond video understanding?

---

> ### Author Response · Authors · 2024-11-21
> **Author Rebuttals[1/2]**
>
> We would like to begin by expressing our sincere gratitude for your thorough review of our paper. We greatly appreciate your suggestions, which are crucial in improving the quality of our paper. The questions you raised are insightful, and we believe we have carefully clarified and addressed them as follows.
>
> > W1&Q1: While the synthetic nature is a strength in terms of scalability and avoiding data leakage, it also limits the ecological validity of the benchmark. Performance on VNBench doesn't guarantee real-world performance. The authors acknowledge this, but a correlation computation between the given benchmark and other real-world ones is needed.
>
> R1&A1: In response to this concern, we add a new section, "cross-task correlation analysis," in Appendix B.2 to provide a detailed correlation computation between VNBench and other real-world benchmarks. Specifically, we evaluated 6 models with varying frame input sizes (16, 32, 48, 64, 96, and 128 frames) as discussed in Section 6. These models were assessed on several real-world video understanding benchmarks, including VideoMME [1], MLVU [2], and LongVideoBench [3].
>
> | #Frames | VNBench | MLVU  | LongVideoBench | VideoMME |
> | ------- | ------- | ----- | -------------- | -------- |
> | 16      | 23.33   | 52.53 | 46.78          | 49.74    |
> | 32      | 29.93   | 54.84 | 47.16          | 49.96    |
> | 48      | 34.15   | 56.22 | 48.75          | 52.81    |
> | 64      | 32.59   | 57.23 | 47.08          | 52.59    |
> | 96      | 37.26   | 60.97 | 48.60          | 53.26    |
> | 128     | 39.70   | 61.44 | 51.40          | 56.11    |
>
> *Table: Evaluation results on VNBench and real-world benchmarks.*
>
> For each benchmark, we calculated the correlation coefficients of the corresponding performance vectors. The results show that all benchmarks achieved correlation coefficients above 0.75. Notably, VNBench demonstrated correlation coefficients higher than 0.9 with MLVU and VideoMME, two commonly used real-world benchmarks. This indicates that VNBench can, to a certain extent, reflect a model's ability to process real-world videos. We show the cross-task correlation matrix below and visualize the correlation matrix in Figure 15.
>
> |                    | **VNBench** | **MLVU** | **LongVideoBench** | **VideoMME** |
> | ------------------ | ----------- | -------- | ------------------ | ------------ |
> | **VNBench**        | 1.000000    | 0.951426 | 0.823635           | 0.912259     |
> | **MLVU**           | 0.951426    | 1.000000 | 0.782742           | 0.897072     |
> | **LongVideoBench** | 0.823635    | 0.782742 | 1.000000           | 0.908773     |
> | **VideoMME**       | 0.912259    | 0.897072 | 0.908773           | 1.000000     |
>
> *Table: Cross-task correlation matrix among VNBench and real-world video understanding tasks.*
>
> [1] Fu, Chaoyou, et al. "Video-mme: The first-ever comprehensive evaluation benchmark of multi-modal llms in video analysis." *arXiv preprint arXiv:2405.21075* (2024).
>
> [2] Zhou, Junjie, et al. "MLVU: A Comprehensive Benchmark for Multi-Task Long Video Understanding." *arXiv preprint arXiv:2406.04264* (2024).
>
> [3] Wu, Haoning, et al. "Longvideobench: A benchmark for long-context interleaved video-language understanding." *arXiv preprint arXiv:2407.15754* (2024).
>
> > W2&Q2: The reliance on CLIP for filtering needle images introduces a potential bias. The performance of the CLIP model itself could influence the results, especially on synthetic videos. A sensitivity analysis or exploration of alternative filtering methods would be valuable.
>
> R2&A2: We conducted a human evaluation on 300 sampled cases and compared the results with those obtained using the CLIP thresholding strategy before we chose to adapt this CLIP-based filtering method, which will be shown in the Appendix of our revised manuscript. Only 2 out of 300 samples showed differing conclusions, demonstrating the effectiveness of the CLIP filtering method.
>
> > W3: Wrong single quotation marks in L18, P1.
>
> R3: Thank you for your suggestion. We have addressed the issue and made the necessary corrections in the revised version.

---

> ### Author Response · Authors · 2024-11-21
> **Author Rebuttals[2/2]**
>
> > Q3: How robust are the results to variations in the types and complexities of the "needles"?
>
> A3: Thank you for your question. In Appendix F.2, we include an additional set of experiments where the needle content is replaced with simpler alternatives, such as fruit images or fixed subtitles. The results, presented in Table 8 and below (SRC for VNBench and NEW for the  alternative one), show a strong correlation between the test results, suggesting that the use of simple and easily recognizable needles ensures the robustness of our results.
>
> | **Video MLLMs**             | Retrieval-E | Retrieval-I-1 | Ordering-E | Ordering-I2 | Counting-E1 |
> | --------------------------- | ----------- | ------------- | ---------- | ----------- | ----------- |
> |                             | SRC/NEW     | SRC/NEW       | SRC/NEW    | SRC/NEW     | SRC/NEW     |
> | **Gemini 1.5 Pro**          | 100.0/100.0 | 96.0/98.0     | 90.7/92.0  | 32.7/31.3   | 60.7/59.3   |
> | **GPT-4o**                  | 100.0/100.0 | 98.0/96.0     | 88.4/90.7  | 45.2/46.0   | 36.8/36.0   |
> | **LLaVA-NeXT-Video-7B**     | 56.7/55.3   | 56.7/60.0     | 0.7/0.7    | 0.7/0.7     | 6.7/3.3     |
> | **ST-LLM**                  | 58.0/59.3   | 64.7/66.7     | 0.0/0.0    | 0.0/0.0     | 21.3/20.6   |
> | **Video-LLaVA-7B**          | 26.0/22.0   | 28.0/26.0     | 0.7/0.7    | 2.0/2.0     | 16.7/15.3   |
> | **Correlation Coefficient** | **0.9990**  | **0.9965**    | **0.9999** | **0.9993**  | **0.9990**  |
>
> Additionally, during the design of VNBench, each task type includes three subcategories corresponding to different needle complexities and types. In these cases, as shown in Table 2, the complexity of the needles introduces some variability in the results. This is because the differences between subcategories are not limited to needle content but also extend to needle formats, which test diverse aspects of video understanding. For instance, subtitle needles assess localized recognition, while image needles evaluate global semantic comprehension. This variability highlights the ability of VNBench to probe different facets of model performance.
>
> > Q4: Could the VideoNIAH framework be adapted to evaluate other multimodal tasks beyond video understanding?
>
> A4: Thank you for your insightful question. The VideoNIAH framework can indeed be adapted to evaluate a broader range of multimodal tasks beyond video understanding. For instance, in high-resolution image understanding, localized edits could be introduced by inserting small image patches as needles. Similarly, in the context of video and audio analysis, a short audio segment could be inserted into a video as a needle to evaluate multimodal integration.
>
> We plan to further explore and extend the applicability of the VideoNIAH framework in future work to enhance its versatility and support a wider range of multimodal evaluation scenarios.

---

> ### Author Response · Authors · 2024-11-26
> **Request for Further Feedback**
>
> Dear Reviewer 9VkG,
>
> We have carefully reviewed your comments and suggestions on our manuscript and have submitted our rebuttal addressing the points raised. We highly value your feedback and would be very grateful if you could take a moment to review our responses.
>
> We sincerely appreciate your time and consideration in providing further feedback.
>
> Best regards,
>
> Authors

---

> ### Author Response · Authors · 2024-12-02
> **Request for Feedback on Rebuttal**
>
> Dear Reviewer 9VkG,
>
> We sincerely appreciate your time and efforts in reviewing our paper.
>
> We have carefully addressed the suggestions and concerns raised by the reviewers. In particular, we have added more detailed revisions based on your feedback, including a robust analysis on needle content, needle type, and sample numbers, as well as a correlation analysis between VNBench and real-world benchmarks.
>
> As the rebuttal deadline approaches, we kindly request that you provide your feedback on the revised version of the paper at your earliest convenience. Your input is invaluable, and we greatly appreciate your timely response.
>
> Thank you once again for your time and thoughtful review. We look forward to hearing from you soon.
>
> Best regards,
>
> Authors

---

### Official Review · Reviewer_ozpA · 2024-11-03

**Soundness:** 3
**Presentation:** 2
**Contribution:** 2
**Rating:** 6
**Confidence:** 3

**Summary:**

This paper proposed a synthetic framework called VideoNIAH to construct the benchmark for evaluating video MLLMs. This framework aims to decouple video content from query-response pairs by corrupting the original videos with "needles" that are irrelevant to the videos.  Based on this frame, a video benchmark named VNBench is further built to evaluate different capabilities of proprietary and open-source video MLLMs, such as temporal perception, chronological ordering, and spatio-temporal coherence.

**Strengths:**

1. The proposed VideoNIAH is a simple and flexible framework for constructing the video benchmark by avoiding expensive human annotation and leveraging existing video datasets.
2. The VNBench seems to be the first synthetic video benchmark for evaluating video MLLMs through three skills including retrieval,
ordering and counting.
3. Comprehensive evaluation experiments are conducted to evaluate close- and open-source MLLMs, further giving some insights for improving the model development and training.

**Weaknesses:**

1. Although the authors mentioned the proposed framework is inspired by language model evaluation. It is still not very clear about the motivation for proposing such a synthetic framework for video benchmarking and why the proposed one is necessary or important.
2. The proposed framework mainly adopts subtitles and unrelated images as the "needle", it is better to discuss whether there are other types of "needle" that can be used.
3. A small number of video samples and the types of selected video datasets are used to build the benchmark. In addition, only three tasks are adopted to evaluate the video MLLMs. It is better to make the benchmark more scalable and diverse with comprehensive evaluation capabilities.

**Questions:**

1. The paper mentioned the used 'needles' are not related to the original videos. Did you conduct the similarity calculation between each video frame and the candidate "needle" before inserting them?
2. Inserting the unrelated visual "needle" may not make the benchmark very challenging. Did you try the "needles" that are similar to the original video frame to a certain extent but are from a different category or situation? Maybe it will make the benchmark more challenging.

---

> ### Author Response · Authors · 2024-11-21
> **Author Rebuttals[1/2]**
>
> We would like to begin by expressing our sincere gratitude for your thorough review of our paper. We greatly appreciate your suggestions, which are crucial in improving the quality of our paper. The questions you raised are insightful, and we believe we have carefully clarified and addressed them as follows.
>
> > W1: Although the authors mentioned the proposed framework is inspired by language model evaluation. It is still not very clear about the motivation for proposing such a synthetic framework for video benchmarking and why the proposed one is necessary or important.
>
> R1: The motivation for proposing VideoNIAH is to address the inefficiencies in evaluating video models during iterative development due to the high cost of constructing datasets and the difficulty in isolating specific skills.
>
> VideoNIAH introduces several key differences compared to previous video benchmarks:
>
> - **Skill-Specific Evaluations**: By inserting multiple spatio-temporal "needles" and generating corresponding query-response pairs based on predefined rules, VideoNIAH enables the precise identification of a model's weaknesses in specific video understanding capabilities. The queries in VideoNIAH focus on specific aspects of video understanding, such as temporal perception, chronological ordering, and spatio-temporal coherence, allowing for more granular evaluation compared to benchmarks that assess general video understanding capabilities.
>
> - **Less Data Leakage**: Synthetic video generation provides more control over the construction of the benchmarks and eliminates the risk of data leakage.
>
> - **Scalability and Efficiency**: VideoNIAH automates the generation of query-response pairs using predefined rules, minimizing manual labor compared to benchmarks that require extensive manual annotation and data filtering. It allows for the use of diverse video sources with flexible lengths, offering significant scalability and efficiency.
>
> > W2: The proposed framework mainly adopts subtitles and unrelated images as the "needle", it is better to discuss whether there are other types of "needle" that can be used.
>
> R2: We include the results of VNBench-Act in Appendix E. VNBench-Act is a synthetic benchmark constructed using the VideoNIAH method. In VNBench-Act, we aim to evaluate the insertion of **short video clips** containing continuous natural frames from Something-Something V2[1]. This ensures that the inserted needle not only carries static image-level semantics but also conveys short-term action meanings.
>
> | **Method**       | **Ret-Act** | **Ord-Act** | **Cnt-Act** |
> | ---------------- | ----------- | ----------- | ----------- |
> | Video-LLaVA      | 27.3        | 0.7         | 7.3         |
> | LLaVA-NeXT-Video | 42.7        | 0.7         | 11.3        |
> | ST-LLM           | 54.7        | 0.0         | 22.7        |
> | GPT-4o           | 85.6        | 11.3        | 10.0        |
>
> *Table: Evaluation results on VNBench-Act, which use short action video clips as needles.*
>
> It is worth noting that when the needles are extended to short video clips, the task becomes more challenging, especially for ordering tasks. This highlights limitations in modeling temporal action sequences effectively.
>
> [1] Goyal, Raghav, et al. "The" something something" video database for learning and evaluating visual common sense." *Proceedings of the IEEE international conference on computer vision*. 2017.

---

> ### Author Response · Authors · 2024-11-21
> **Author Rebuttals[2/2]**
>
> > W3: A small number of video samples and the types of selected video datasets are used to build the benchmark. In addition, only three tasks are adopted to evaluate the video MLLMs. It is better to make the benchmark more scalable and diverse with comprehensive evaluation capabilities.
>
> R3:
>
> Thank you for your constructive feedback. As detailed in Appendix F.3, we double the number of test samples using the VideoNIAH framework. We evaluated the impact of the increased sample size on the average accuracy across the three VNBench tasks: Retrieval, Ordering, and Counting. Our analysis in Table 9 and below (SRC for VNBench and MORE for the  expanded one) indicates a relatively high correlation coefficient between the results of the original dataset and the enlarged benchmark, demonstrating the robustness of our benchmark with respect to the current sample size.
>
> | **Video MLLMs**             | Retrieval  | Ordering   | Counting   | **Overall (src)** |
> | --------------------------- | ---------- | ---------- | ---------- | ----------------- |
> |                             | SRC/MORE   | SRC/MORE   | SRC/MORE   | SRC/MORE          |
> | Gemini 1.5 Pro              | 90.7/89.2  | 72.9/73.2  | 36.7/35.7  | 66.7/66.0         |
> | GPT-4o                      | 95.3/96.7  | 73.4/75.2  | 24.5/24.3  | 64.4/65.4         |
> | LLaVA-NeXT-Video-7B         | 44.2/42.4  | 0.4/2.3    | 15.5/13.5  | 20.1/19.4         |
> | ST-LLM                      | 51.3/49.4  | 0.0/0.5    | 16.7/15.7  | 22.7/21.9         |
> | Video-LLaVA-7B              | 23.8/23.8  | 1.1/2.6    | 12.4/11.1  | 12.4/12.5         |
> | **Correlation Coefficient** | **0.9998** | **0.9577** | **0.9990** | **0.9996**        |
>
> Additionally, we appreciate your suggestion regarding diversity. In future work, we plan to further expand the types of tasks included in VNBench to enhance its comprehensiveness and evaluation capabilities.
>
> > Q1: The paper mentioned the used 'needles' are not related to the original videos. Did you conduct the similarity calculation between each video frame and the candidate "needle" before inserting them?
>
> A1: As mentioned in Section 3.3, we utilized the CLIP model to calculate the similarity between the inserted needle and all frames of the video haystack, and applied a threshold-based strategy to filter relevant samples. Additionally, we conducted a human evaluation on 300 sampled cases and compared the results with those obtained using the CLIP thresholding strategy, which will be shown in the Appendix of our revised manuscript. Only 2 out of 300 samples showed differing conclusions, demonstrating the effectiveness of the CLIP filtering method.
>
> > Q2: Inserting the unrelated visual "needle" may not make the benchmark very challenging. Did you try the "needles" that are similar to the original video frame to a certain extent but are from a different category or situation? Maybe it will make the benchmark more challenging.
>
> A2: We appreciate the reviewer’s suggestion, but we would like to clarify a few points.
>
> First, the goal of VideoNIAH is to decouple image recognition from video-specific temporal abilities. For this reason, we specifically chose unrelated "needles" to evaluate a model’s targeted video capabilities without introducing additional complexity related to visual similarity.
>
> Second, the benchmark was designed with graded difficulty within tasks and variability across task types, which ensures that it remains challenging. Even state-of-the-art closed-source models, such as GPT-4 and Gemini 1.5 Pro, do not achieve perfect performance, further highlighting the benchmark’s challenge.
>
> Furthermore, inserting similar video frames could indeed increase the task difficulty by making the target distinction harder. However, this would complicate our ability to isolate and explore the specific video understanding capabilities we aim to assess. In this case, the performance degradation on the benchmark would result from more complex factors, making it less effective for evaluating the targeted abilities.

---

> ### Comment · Reviewer_ozpA · 2024-11-25
>
> Thank you for your response. My concerns have been addressed. I'm willing to raise my rating to 6.

---

> > ### Author Response · Authors · 2024-11-27
> >
> > Dear Reviewer ozpA,
> >
> > We would like to sincerely thank you for the positive feedback and the higher rating you provided. We greatly appreciate the time and effort you put into reviewing my work.
> >
> > Thank you again for your thoughtful review!
> >
> > Best regards,
> >
> > Authors

---

### Official Review · Reviewer_FQGX · 2024-11-04

**Soundness:** 3
**Presentation:** 3
**Contribution:** 3
**Rating:** 6
**Confidence:** 3

**Summary:**

This paper introduces Video Needle-In-a-Haystack (VideoNIAH), a framework for synthetic video generation that utilizes intra-frame and inter-frame edits to create specific modifications within video segments. These modifications, referred to as "needles," are used to assess the capabilities of video MLLMs in handling challenging video content. Additionally, the authors present VNBench, a benchmark compiled from these synthetic videos, which evaluates video MLLMs across three video understanding tasks: retrieval, ordering, and counting. The benchmark is tested on 12 MLLMs, both proprietary and open-source.

**Strengths:**

- The paper presents a framework to create a synthetic (i.e.: edited video) video that can be automated based on certain predetermined rules and tasks. These edits are posed as “needles” in a “haystack” (the original video segment), and the query for the MLLM is based on the inserted needles. The proposed method is easily scalable given the minimal labeling/definition process that needs to be set manually.
- The curated VNBench benchmark can be used for evaluation of MLLM in video understanding, primarily spatial and temporal understanding. This is presented through tasks of retrieval, counting and ordering based queries from the curated video benchmark.
- The paper evaluates 12 MLLMs on the proposed benchmark with extensive evaluation on the “needle” placements and difficulty.

**Weaknesses:**

- The rule-based automation process for dataset curation is unclear. For example, Figure 1(b) lacks clarity in illustrating this process, which is essential to understanding the scalability benefits of this approach, which can be improved.
- While the method is scalable and captures the temporal dependency well, the scope of video understanding through this framework seems limited. For instance, the ability of the model in understanding actions, fine-grained action recognition cannot be ascertained, especially since MLLMs are used in this context extensively.
- Given the nature of the benchmark, the sampling strategy of the video MLLM I believe plays a significant role in its performance. For instance, if any of the frames with a “needle” is not sampled, the model fails to correctly answer that query. Clarification on how this issue is addressed, or suggested solutions, would strengthen the framework’s reliability.
- Can the authors explain if each cell in Figure 8 corresponds to a single test sample? It would be better to enhance this image with a heat map to better understand how the depth of the needle impacts the MLLMs performance. In addition, is there a reason that a depth higher than 10% was not considered?
- Section 6: The mention of the "training recipe" lacks context on what is being trained, creating some confusion. A clearer introduction to this section would aid comprehension.

**Questions:**

- **Counting E2 Task:** What factors might explain the poor performance of Gemini and GPT-4 on the counting E2 task (Figure 5)?
- **Consistency Check:** Could the authors verify the data in Figure 5 and Table 2? For instance, Video-LLaVa and VideoChat2 seem to perform better than Gemini on the counting E2 task in Figure 5, but Table 2 suggests otherwise.
- **Haystack Length Variation:** In Section 5, where the "Effect of Needle Position" is discussed, the authors mention that the haystack is fixed. Could the authors clarify how the length of the haystack is adjusted in this scenario?

---

> ### Author Response · Authors · 2024-11-21
> **Author Rebuttals[1/2]**
>
> We would like to begin by expressing our sincere gratitude for your thorough review of our paper. We greatly appreciate your suggestions, which are crucial in improving the quality of our paper. The questions you raised are insightful, and we believe we have carefully clarified and addressed them as follows.
>
> > **W1:** The rule-based automation process for dataset curation is unclear. For example, Figure 1(b) lacks clarity in illustrating this process, which is essential to understanding the scalability benefits of this approach, which can be improved.
>
> **R1:** The rule-based automation process involves designing query templates based on the **task type and needle type**. For example, for a task of the 'order' type and a needle of the 'subtitle' type, the query might ask, 'What is the order of subtitles appearing in the video?' The corresponding options and ground truth are directly determined by how we insert the needles and their content, which is known in advance.
>
> Since the questions depend solely on the type and method of needle insertion, generating query-response pairs can be fully automated based on predefined rules. We update Figure 1(b) to include some example rule types. However, we recommend readers refer to Appendix A.4 and Figure 13 for more details on the rules to generate queries and responses.
>
> > **W2:** While the method is scalable and captures the temporal dependency well, the scope of video understanding through this framework seems limited.  For example, understanding actions.
>
> R2: We include the results of VNBench-Act in Appendix E. VNBench-Act is a synthetic benchmark constructed using the VideoNIAH method. In VNBench-Act, we aim to evaluate the insertion of **short action video clips** containing continuous natural frames from Something-Something V2[1]. This ensures that the inserted needle not only carries static image-level semantics but also conveys short-term action meanings.
>
> | **Method**       | **Ret-Act** | **Ord-Act** | **Cnt-Act** |
> | ---------------- | ----------- | ----------- | ----------- |
> | Video-LLaVA      | 27.3        | 0.7         | 7.3         |
> | LLaVA-NeXT-Video | 42.7        | 0.7         | 11.3        |
> | ST-LLM           | 54.7        | 0.0         | 22.7        |
> | GPT-4o           | 85.6        | 11.3        | 10.0        |
>
> *Table: Evaluation results on VNBench-Act, which use short action video clips as needles.*
>
> It is worth noting that when the needles are extended to short video clips, the task becomes more challenging, especially for ordering tasks. This highlights limitations in modeling temporal action sequences effectively.
>
> [1] Goyal, Raghav, et al. "The" something something" video database for learning and evaluating visual common sense." *Proceedings of the IEEE international conference on computer vision*. 2017.
>
> > **W3:** Given the nature of the benchmark, the sampling strategy of the video MLLM I believe plays a significant role in its performance. Clarification on how this issue is addressed, or suggested solutions, would strengthen the framework’s reliability.
>
> **R3:** Thank you for your insightful comment. We acknowledge that the sampling strategy of the video MLLM can indeed influence performance. To address this, we set the interval for inserting the needle to 1 second, ensuring that if the model samples at 1 frame per second (fps), it will capture the needle in the input.
>
> Currently, some models (e.g. Video-LLaVA, VideoChat2, ST-LLM, .etc) do not operate at 1fps, primarily due to computational limitations, which is a inherent shortcoming for achieving better video understanding ability. Dense sampling is fundamental for comprehensive video understanding, while sparse sampling can lead to the neglect of crucial video details.
>
> On the other hand, we observe that current state-of-the-art models, such as GPT-4 and Gemini 1.5 Pro, operate at 1fps, which means no needles are missed during VNBench evaluation. However, these models still struggle to achieve optimal performance, suggesting advanced improvement on video understanding in future.

---

> ### Author Response · Authors · 2024-11-21
> **Author Rebuttals[2/2]**
>
> > **W4:** Can the authors explain if each cell in Figure 8 corresponds to a single test sample? It would be better to enhance this image with a heat map to better understand how the depth of the needle impacts the MLLMs performance. In addition, is there a reason that a depth higher than 10% was not considered?
>
> **R4:** Thank you for your valuable suggestions. In the original figure, each cell indeed corresponds to a single test sample. We revise Figure 8 into heatmaps to show the modal ability more accurately. Now, we construct 32 samples, each corresponding to needle depths ranging from 0% to 90% (at 10% intervals) and haystack durations from 10s to 180s (at 5s intervals) with the same needle content. For each cell position, we report the average accuracy across the 32 samples.
>
> Additionally, the depth range in the original figure was incorrectly labeled as 0-9; the correct range is from 0% to 90%, with intervals of 10%.
>
> > **W5:** Section 6: The mention of the "training recipe" lacks context on what is being trained, creating some confusion. A clearer introduction to this section would aid comprehension.
>
> **R5:** We have provided details of the training process in the appendix F, including the model architecture, training data, and training hyperparameters. We also add more details in appendix F, including input format and  loss functions.
>
> > **Q1:** **Counting E2 Task:** What factors might explain the poor performance of Gemini and GPT-4 on the counting E2 task (Figure 5)?
>
> **A1:**  The poor performance of Gemini and GPT-4 on the Counting E2 task can be attributed to two main factors related to the task's requirements for both temporal and spatial co-occurrence:
>
> 1. **Poor temporal co-occurrence understanding**: Both Gemini 1.5 Pro and GPT-4 performed suboptimally on other tasks that only require temporal co-occurrence awareness, such as Counting-E1 and Counting-I. This suggests that the models may struggle with tracking temporal relationships effectively.
> 2. **Weak spatial co-occurrence perception**: We conduct a simple experiment to evaluate GPT-4o's ability to perceive spatial co-occurrence. Specifically, we select an image and randomly attached 0 to 4 smaller fruit images to different regions, which can be regarded as a image-level NIAH test. We then asked GPT-4o to predict the number of attached fruit images on 100 curated samples, and it achieved an accuracy of 72/100. In the Counting E2 task, where 4 different clips in the video sequence each have image needles (needle number randomly choice from 0 to 4 per clip) attached, the accumulated accuracy would be 0.72^4≈0.26. This probabilistic estimation indicates that spatial co-occurrence understanding might also be a limiting factor for these models on this task.
>
> > **Q2:** **Consistency Check:** Could the authors verify the data in Figure 5 and Table 2? For instance, Video-LLaVa and VideoChat2 seem to perform better than Gemini on the counting E2 task in Figure 5, but Table 2 suggests otherwise.
>
> **A2:** We appreciate the reviewer’s suggestions. During the plotting process, a small portion of the data (specifically, counting-edit-level2 and counting-insert for certain models, e.g., Video-LLaVA, VideoChat2) was mistakenly mixed. We correct the results in Figure 5 of the manuscript. Additionally, all numerical results were already provided in Table 11 of Appendix G. Thanks for you advice!
>
> > **Q3:** **Haystack Length Variation:** In Section 5, where the "Effect of Needle Position" is discussed, the authors mention that the haystack is fixed. Could the authors clarify how the length of the haystack is adjusted in this scenario?
>
> **A3:** We selected a fixed video segment and randomly cropped it to obtain haystacks of varying lengths. For the position test, we used the same cropped video segment for haystacks of the same length, altering only the depth of inserted needle .

---

> > ### Comment · Reviewer_FQGX · 2024-11-23
> > **Prevailing concerns**
> >
> > Thank you for the authors in the comprehensive review response. Few of my concerns still remain unresolved.
> > - In regard to R1: While Appendix A.4 and Figure 13 clarify the dataset curation process with sample examples, I have the following questions, particularly regarding replicating or expanding a similar dataset—a key contribution of the paper:
> >     - Given a task and a needle type, is there a single query to extract relevant data? If not, how are the queries automatically formulated?
> >     - What specific rules are used? While it is mentioned that the rules are shown in Figure 1(b), this remains unclear.
> >
> > - R3: I believe this partially solves the problem. While dense sampling is indeed necessary for improved video understanding, there exists the difficulty in effectively evaluating all models under the same setting "fairly" (since the sampling strategy is a model-specific design choice, it is not inherently unfair; however, the comparison does not occur on an entirely level playing field).However, the scalable nature of the proposed approach is noteworthy and opens avenues for further improvement.
> >
> > - A1-- Performance of GPT4o and Gemini:
> >     - Both models seem to perform better than majority of other models on the mentioned tasks; Counting-E1 (60.7 & 36.8) and Counting-I (42 & 36.2), which are not suboptimal. This actually shows the model capabilities in temporal understanding.
> >     - However, without comparing other models on a similar test, this observation is difficult to fully validate. The performance drop to 0% on Counting-E2 for GPT-4 is especially puzzling, given its apparent advantage in sampling strategies over other models. Further clarification on this discrepancy would be valuable.
> >
> > - A2: Figure 5
> >     - Figure 5 contains a few inconsistencies that should be addressed. For example, some plots are missing models that could provide a more comprehensive view. Additionally, for Counting-E2, the reported value for LLaVA-One-Vision-72B in the table is 14.7%, while an approximation from the plot would be around 25%. Ensuring consistency between plots and tables is essential.

---

> > > ### Author Response · Authors · 2024-11-23
> > > **Response to the prevailing concerns[1/2]**
> > >
> > > ​
> > >
> > > We sincerely appreciate your careful review, which has been immensely helpful in improving our paper!
> > >
> > > > Q-I: Given a task and a needle type, is there a single query to extract relevant data? If not, how are the queries automatically formulated? What specific rules are used? While it is mentioned that the rules are shown in Figure 1(b), this remains unclear.
> > >
> > > A-I:
> > >
> > > - When we extract a task and a needle type, we directly collect needle contents from a human-chosen set, which may be subtitles, images from a special category, .etc. These needle contents represent the different level sub-task under a specific task. Additionally, **the query is formulated with needle type and task type by human-defined rule before data construction, and is kept same in the whole sub-task data.**
> > > - We directly use human-defined query templates as rules. For example, in counting task, the rule is "How many {needle type} appears in the video?", and for ordering task, the rule is "What is the order of {needle type} that appears in the video?". These rule is influenced by needle type and task type only. And the query generated by these rule is kept same in a single sub-task, since the needle type and task type in a sub-task is also unchanged. For 9 sub-tasks in VNBench, the specific queries can be referred in the task examples in Appendix I.
> > >
> > > > Q-II: I believe this partially solves the problem. While dense sampling is indeed necessary for improved video understanding, there exists the difficulty in effectively evaluating all models under the same setting "fairly" . However, the scalable nature of the proposed approach is noteworthy and opens avenues for further improvement.
> > >
> > > A-II:
> > >
> > > - As you rightly pointed out, the sampling strategy is an integral part of model design. Additionally, the models we evaluated exhibit significant gaps in training data and pre-trained weights, making a perfectly fair comparison inherently challenging.
> > >
> > > - Nevertheless, we appreciate for your valuable feedback, we consider the scalability of our approach. Looking ahead, we believe that incorporating **different needle durations** could serve as a valuable dimension for more comprehensive evaluations. The longer the needle duration, the more likely the needle is to be sampled.
> > >
> > > > Q-III: Performance of GPT4o and Gemini:Both models seem to perform better than majority of other models on the mentioned tasks, which actually shows the model capabilities in temporal understanding. However, without comparing other models on a similar test, this observation is difficult to fully validate.
> > >
> > > A-III:
> > >
> > > - All models were tested on the same evaluation set. We guess that your concern may stem from the fact that some models were not evaluated in 1-fps setting. We acknowledge this possibility. However, it is worth noting that the open-sourced model LLaMA-VID was also evaluated under the 1-fps setting and show poor performance. Additionally, many models (e.g., Video-LLaVA, Video-Chat2, ST-LLM) experience performance degradation when longer frame inputs are used during inference.
> > >
> > > - Furthermore, the experimental results presented in Figure 11 (on the counting task) demonstrate that, for the same series of models, **simply increasing the number of input frames contributes less when the frame count becomes sufficiently large** (>32, in our results). This suggests that the high performance of GPT-4 and Gemini 1.5 Pro is not solely due to the frame sampling strategy, but rather reflects their inherent capabilities in temporal understanding.

---

> ### Author Response · Authors · 2024-11-23
> **Response to the prevailing concerns[2/2]**
>
> > Q-IV: The performance drop to 0% on Counting-E2 for GPT-4 is especially puzzling, given its apparent advantage in sampling strategies over other models. Further clarification on this discrepancy would be valuable.
>
> A-IV:
>
> Thank you for your insightful comment. The poor performance of GPT-4 on the Counting-E2 task indeed worths further investigation.
>
> - In our experiments, the input prompts were kept consistent across all models. For both the GPT-4 series and Gemini 1.5 Pro, we applied the same frame sampling strategy and query format. Therefore, we believe there were no issues in the testing procedure that could have influenced the results.
>
> - Interestingly, we found that all models performed poorly on Counting-E2, with even the best model (LLaVA-OneVision-72B) achieving only 14.6% accuracy on this task, which is lower than best accuracy on other sub-tasks. While counting is relatively easy for humans, it proves more challenging for multimodal large language models at the spatial level, as noted by Molmo[1]. We will also conduct more comprehensive experiments and analysis on the counting ability in video MLLMs base on the VideoNIAH framework.
>
> [1] Deitke, Matt, et al. "Molmo and pixmo: Open weights and open data for state-of-the-art multimodal models." *arXiv preprint arXiv:2409.17146* (2024).
>
> > Q-V: Figure 5 contains a few inconsistencies that should be addressed. For example, some plots are missing models that could provide a more comprehensive view.  Ensuring consistency between plots and tables is essential.
>
> A-V: Apologized for the mistakes! We modify Figure 5 in the revised version. And we also conclude **all model's performance** in Table 11 and Figure 5. Once again, thank you for your thorough review, which has been a great help in improving our paper.

---

> > ### Comment · Reviewer_FQGX · 2024-11-24
> > **Response to rebuttal**
> >
> > Thank you for responding to the queries raised.
> >
> > Q-III on the performance of Gemini-1.5 and GPT4 was regarding the original author response A1-1, which stated the Gemini and GPT4 models were performing sub-optimally on temporal coherence. However, this is not the case as they perform better than majority of other models on the mentioned tasks; Counting-E1 (60.7 & 36.8) and Counting-I (42 & 36.2). Secondly, the "test" referred to, is the "**Weak spatial co-occurrence perception**" conducted by the authors and reported in the initial response A1-2. Without a comparison of the performance of other models (since only GPT4 is evaluated on this test), the "*weak spatial co-occurence of perception*" as a reason for the sub-optimal performance in Counting-E2 is not well supported.
> >
> > However the authors do mention in response A-IV that MLLMs find counting a challenging task and one that needs exploration. The surprising case is that GPT4 underperforms in comparison other open MLLMs such as VideoChatGPT, Video-LLaMA and LLaMA-Vid, which would be interesting to understand.

---

> > > ### Author Response · Authors · 2024-11-25
> > > **Response to 'Response to rebuttal' [1/2]**
> > >
> > > We appreciate the reviewer's in-depth discussion.
> > >
> > > Based on the reviewer’s suggestion, we test the open-source models (LLaVA-OneVision series) and the proprietary model (Gemini 1.5 Pro) on our designed "Weak spatial co-occurrence perception" experiments. In fact, these models exhibited performance similar to GPT-4o (accuracy ranging from 65% to 82%), which suggests that we may need to consider the cause from another perspective.
> > >
> > > We must acknowledge that performing counting simultaneously across both spatial and temporal dimensions is a challenging task for VideoMLLMs, with the highest performance being only 14.7%. In contrast, for other sub-tasks, the SoTA models achieve an accuracy of over 40%, which is sufficient to reflect their abilities in these aspects.
> > >
> > > Since our tasks are in a multi-choice QA format, and most video MLLMs (excluding VideoChatGPT and VideoLLaMA2, which directly output sentence and must use GPT to evaluate) have strong instruction-following capabilities, their output is constrained to the four provided option letters. This means that even if the model cannot select the correct answer, it will still output one of the option letters.
> > >
> > > As mentioned in [1] and [2], LLMs inherently have preferences for certain options, and different models show varying sensitivity to the order of options. When options are shuffled (in our circular evaluation) and a model lacks the ability to analyze the question but is forced to output an answer, it tends to rely on internal biases to select its preferred option letter rather than making a choice based on option content. This can lead to situations where the model outputs the same option for a shuffled version of the question answer candidates, even though the content corresponding to that option differs.
> > >
> > > We analyzed this phenomenon on 9 sub-tasks in **Table 12, 13, 14**. And here we show the result on counting sub-tasks which are with relatively low SoTA performance. First, we examined the option consistency of LLaVA-OneVision series, Gemini 1.5 Pro, and GPT-4 series. Since our circular evaluation involves four iterations with shuffled option orders, only predictions where the option content remains the same across all four iterations are considered consistent. We define **Consist.** as the proportion of consistent predictions across these four iterations in all test samples, and **Acc.** as the accuracy of the model's predictions under circular evaluation.
> > >
> > > In this setup, two outcomes are possible:
> > >
> > > 1. The predictions are consistent, meaning the model may predict the same answer as the GroundTruth (correct) or a different answer (incorrect).
> > > 2. The predictions are inconsistent, which results in the answer being marked as incorrect in circular evaluation.
> > >
> > > Thus, accurate predictions are a subset of consistent predictions. We compute the conditional probability **Acc.|Consist.**, which represents the accuracy of the model’s prediction given that the results are consistent.

---

> ### Author Response · Authors · 2024-11-25
> **Response to 'Response to rebuttal' [2/2]**
>
> | **Video MLLMs**      | **Counting-E1** |          |                    | **Counting-E2** |          |                    | **Counting-I** |          |                    |
> | -------------------- | --------------- | -------- | ------------------ | --------------- | -------- | ------------------ | -------------- | -------- | ------------------ |
> |                      | Acc.            | Consist. | **Acc.\|Consist.** | Acc.            | Consist. | **Acc.\|Consist.** | Acc.           | Consist. | **Acc.\|Consist.** |
> |                      |                 |          |                    |                 |          |                    |                |          |                    |
> | random choice        | -               | -        | **25.0**           | -               | -        | **25.0**           | -              | -        | **25.0**           |
> | Gemini 1.5 Pro       | 60.7            | 67.3     | **90.2**           | 7.3             | 27.3     | **26.7**           | 42.0           | 54.0     | **77.8**           |
> | GPT-4o               | 36.8            | 42.9     | **85.8**           | 0.0             | 5.3      | **0.0**            | 36.1           | 55.0     | **72.2**           |
> | GPT-4-turbo          | 37.6            | 44.9     | **83.7**           | 0.0             | 3.6      | **0.0**            | 32.4           | 36.2     | **89.5**           |
> | LLaVA-OneVision-0.5B | 6.7             | 33.3     | **20.1**           | 9.3             | 26.0     | **35.7**           | 18.7           | 73.3     | **25.5**           |
> | LLaVA-OneVision-7B   | 41.3            | 72.0     | **57.3**           | 8.7             | 48.7     | **17.9**           | 27.3           | 92.0     | **29.6**           |
> | LLaVA-OneVision-72B  | 42.7            | 60.7     | **70.3**           | 14.7            | 60.7     | **24.2**           | 30.7           | 80.0     | **38.4**           |
>
>
>
> Our conclusions are as follows:
>
> 1. When the conditional probability **Acc.|Consist.** is significantly higher than 25% (roughly equivalent to random guessing), the accuracy can effectively reflect the model's ability to handle the task. In **Counting-E1** and **Counting-I**, the results from the model performance comparison are valuable.
> 2. When the conditional probability **Acc.|Consist.** fluctuates around 25% or falls below 25%, comparisons of accuracy have limited significance. In this case, the conclusion is that the models are unable to solve the task, meaning that all models perform unsatisfactorily on **Counting-E2**. This represents a common issue across all models and indicates an area where VideoMLLMs need optimization.
> 3. For GPT-series models, we find that compared to Gemini 1.5 Pro and LLaVA-OneVision series models, their **Consistency Rate** on unsolvable tasks (e.g., **Counting-E2**) is very low (5.3% for GPT-4o, 3.6% for GPT-4-turbo). This suggests that these models are highly sensitive to the order of options, resulting in a 0.0% accuracy in circular evaluation. While other models also fail to provide correct answers (reflected by the conditional probability **Acc.|Consist.** around 25%, roughly equivalent to random guessing), their higher **Consistency Rate** leads to slightly higher **Acc.** values. However, as concluded in point 2, such comparisons are not significant. This phenomenon can also be validated in Table 7, where GPT-4o shows performance closer to Gemini 1.5 Pro with fewer evaluation iterations.
> 4. The low **Consistency** performance of GPT-series models is seen only in tasks they cannot solve. On tasks they can handle, their **Consistency Rate** is high (in Table 12, 13).
>
> The above conclusions suggest that for VNBench, comparisons at high **Acc.** levels reflect the models' abilities, while low **Acc.** levels indicate a common issue for all Video MLLMs. In fact, this conclusion holds true for most multi-choice QA benchmarks. Meanwhile, inspired by the reviewer's advice, we believe that **Acc.|Consist.** may also be an important metric in circular evaluation.
>
> [1] Wei, Sheng-Lun, et al. "Unveiling Selection Biases: Exploring Order and Token Sensitivity in Large Language Models." *arXiv preprint arXiv:2406.03009* (2024).
>
> [2]  Wang, Haochun, et al. "Beyond the answers: Reviewing the rationality of multiple choice question answering for the evaluation of large language models." *arXiv preprint arXiv:2402.01349* (2024).

---

> > ### Comment · Reviewer_FQGX · 2024-11-25
> >
> > Thank you for the authors comprehensive analysis on this, which shows better insights on this evaluation setting.
> >
> > My main concerns have been well addressed. While the sampling strategy of the MLLM is impacted by the evaluation setting, I believe the scalable nature of the synthetic benchmark is promising for broader applications. I will keep my score at the current rating.

---

> > > ### Author Response · Authors · 2024-11-27
> > >
> > > Dear Reviewer FQGX,
> > >
> > > We truly appreciate your attention to detail and the expertise you brought to the review process.
> > >
> > > Thank you once again for your support!
> > >
> > > Best regards,
> > >
> > > Authors

---

### Official Review · Reviewer_aagQ · 2024-11-04

**Soundness:** 3
**Presentation:** 3
**Contribution:** 3
**Rating:** 6
**Confidence:** 3

**Summary:**

This paper proposes a scalable synthetic framework for benchmarking video MLLMs. It decouples video content from query-response pairs and separates different aspects of video understanding skills. The authors construct a synthetic video benchmark with tasks like retrieval, ordering, and counting to evaluate video models' temporal perception, chronological order, and spatio-temporal coherence. The experiment on VNBench reveals performance gaps and weaknesses in current video models.

**Strengths:**

1. VideoNIAH offers a novel approach to evaluating video models by decoupling video content and query-response pairs.

2. VNBench provides a scalable and flexible way to evaluate video understanding across different dimensions and video lengths, overcoming some limitations of traditional benchmarks.

3. The paper conducts a thorough evaluation of multiple models, analyzing various factors such as haystack length, needle number, and model settings, providing valuable insights for model improvement.

**Weaknesses:**

1. While VNBench assesses important aspects of video understanding, it may not cover all possible video-related tasks and scenarios.

2. Despite the relatively low cost of the evaluation method proposed in this paper, it is vulnerable to being gamed by models. Models may develop strategies to perform well on the tasks without truly understanding the video content, thus compromising the reliability of the evaluation.

3. The overall tasks in the proposed evaluation framework are relatively simplistic. After targeted training, models can achieve high scores with relative ease, which may not accurately reflect their ability to handle more complex and diverse real-world video understanding scenarios. This simplicity could lead to an overestimation of the models' capabilities and may not provide a comprehensive assessment of their true understanding and generalization ability in practical applications.

**Questions:**

Please see weakness.

---

> ### Author Response · Authors · 2024-11-21
> **Author Rebuttals**
>
> We would like to begin by expressing our sincere gratitude for your thorough review of our paper. We greatly appreciate your suggestions, which are crucial in improving the quality of our paper. The questions you raised are insightful, and we believe we have carefully clarified and addressed them as follows.
>
> > **W1:**  While VNBench assesses important aspects of video understanding, it may not cover all possible video-related tasks and scenarios.
>
> **R1:**  As stated in our paper, we acknowledge that VNBench is not a fully comprehensive benchmark. The goal of the proposed VideoNIAH method is to decompose the capabilities of video understanding models, enabling focused evaluation of specific abilities, such as temporal ordering (ordering task) or spatio-temporal coherence perception (counting task). By decoupling different aspects of video understanding, we aim to better identify the particular shortcomings of existing models.
>
> On the other hand, the three sub-tasks of VNBench (retrieval, ordering, and counting) align with key capabilities in current video understanding. This is further supported by the strong correlation between VNBench evaluation results and those of several real-world benchmarks. In Appendix B.2, we provide a detailed analysis of VNBench’s correlation with real-world video benchmarks VideoMME[1], MLVU[2], and LongVideoBench[3]. Our results show that VNBench closely aligns with the evaluation of these real-world benchmarks. Moreover, the performance across VNBench’s three sub-tasks offers valuable insights into where models may be lacking in specific abilities.
>
> |                    | **VNBench** | **MLVU** | **LongVideoBench** | **VideoMME** |
> | ------------------ | ----------- | -------- | ------------------ | ------------ |
> | **VNBench**        | 1.000000    | 0.951426 | 0.823635           | 0.912259     |
> | **MLVU**           | 0.951426    | 1.000000 | 0.782742           | 0.897072     |
> | **LongVideoBench** | 0.823635    | 0.782742 | 1.000000           | 0.908773     |
> | **VideoMME**       | 0.912259    | 0.897072 | 0.908773           | 1.000000     |
>
> *Table: Cross-task correlation matrix among VNBench and real-world video understanding tasks.*
>
> [1] Fu, Chaoyou, et al. "Video-mme: The first-ever comprehensive evaluation benchmark of multi-modal llms in video analysis." *arXiv preprint arXiv:2405.21075* (2024).
>
> [2] Zhou, Junjie, et al. "MLVU: A Comprehensive Benchmark for Multi-Task Long Video Understanding." *arXiv preprint arXiv:2406.04264* (2024).
>
> [3] Wu, Haoning, et al. "Longvideobench: A benchmark for long-context interleaved video-language understanding." *arXiv preprint arXiv:2407.15754* (2024).
>
> > **W2&W3:** The overall tasks in the proposed evaluation framework are relatively simplistic and with low cost. Models may develop strategies to perform well on the tasks without truly understanding the video content, thus compromising the reliability of the evaluation. After targeted training, models can achieve high scores with relative ease, which may not accurately reflect their ability to handle more complex and diverse real-world video understanding scenarios.
>
> **R2:** First, the goal of the proposed VideoNIAH method is to assess specific aspects of a model’s capabilities,  as a tool for analyzing strengths and weaknesses. We do not encourage models to simply "optimize" for scores on distribution-matching data.
>
> Additionally, VNBench is categorized according to different capability dimensions, allowing us to focus on evaluating particular aspects we aim to explore, such as temporal and spatio-temporal coherence. The tasks in VNBench are designed to capture what we consider essential capabilities for current video understanding. As a result, **VNBench exhibits a relatively low bias towards real-world scenarios**, as shown in the correlation analysis in **R1**.
>
> Moreover, all benchmarks, including VNBench, have inherent biases. Typically, when training large multimodal models, we do not rely on a single benchmark metric as the evaluation criterion. Instead, we adopt a **multi-benchmark approach** for comprehensive evaluation. This means that VNBench is generally used in conjunction with other benchmarks to provide a more accurate evaluation of a model’s true capabilities. The strength of VNBench lies in its ability to **decouple specific model capabilities**, enabling targeted optimization of particular skills.

---

> ### Comment · Reviewer_aagQ · 2024-11-25
>
> Thank the author for the rebuttals. My main concerns have been addressed, so I maintain my score as 6. Looking forward for your open-sourced codebase and datasets.

---

> > ### Author Response · Authors · 2024-11-27
> >
> > Dear Reviewer aagQ,
> >
> > We truly appreciate your attention to detail and the expertise you brought to the review process.
> >
> > Thank you once again for your support!
> >
> > Best regards,
> >
> > Authors

---

### Author Response · Authors · 2024-11-21
**General Response of Revisions**

Dear Reviewers,

Thank you for your valuable feedback and insightful suggestions. We have carefully considered your comments and made the following revisions to the manuscript:

1. **Figure 1(b)**: We have added rule examples to improve clarity and better illustrate the concept.
2. **Figure 5**: We have revised the figure to address discrepancies in the data, ensuring that the visual representation aligns with the updated results.
3. **Figure 8**: We have used more position test examples and replaced the binary visualization with a heatmap for better interpretability.
4. **Appendix B.2**: We have added a cross-task correlation analysis, which demonstrates the correlation among VNBench and real-world video benchmarks.
5. **Appendix E**: We have included the evaluation results on VNBench-ACT, which utilizes short video clips as needles to further investigate task performance.
6. **Appendix F.2**: A robustness analysis on needle content has been added to assess the impact of varying needle content on the results.
7. **Appendix F.3**: We have added a robustness analysis on sample number, evaluating the effect of different sample sizes on the performance of the models.
8. **Appendix G**: We have add more details of training recipe.
8. **Appendix J**: We have add consistency evaluation of VNBench testing metric.

We believe these revisions have significantly improved the clarity and comprehensiveness of the manuscript, and we hope that the changes meet your expectations.

---

### Comment · Area_Chair_5YrU · 2024-12-03
**Discussion due soon**

Dear all reviewers,

Our reviewer-author discussion will end soon. For each of you, please check all the files and see if anything you'd like to discuss with authors.

Best,
Your AC

---

### Meta-Review · Area_Chair_5YrU · 2024-12-18

**Metareview:**

This paper proposes a benchmark construction framework through synthetic video generation. A VNBench is developed to include tasks like retrieval, ordering, and counting to evaluate video understanding. The reviewers mostly favor the proposed method, with [9VkG] indicating the real-world gap, clip bias, and technical presentation issues. In the discussion phase, the authors provide more detailed experiments and analysis. Overall, the AC has checked all the files and feels the authors have sufficiently addressed the raised issues. These revisions shall be incorporated into the camera-ready version.

**Additional Comments On Reviewer Discussion:**

[9VkG] indicating the real-world gap, clip bias, and technical presentation issues. In the discussion phase, the authors provide more detailed experiments and analysis, which addresses the raised issues.

---

### Decision · Program_Chairs · 2025-01-22

Accept (Poster)